# Length-dependent motions of SARS-CoV-2 frameshifting RNA pseudoknot and alternative conformations suggest avenues for frameshifting suppression

Shuting Yan[1,5], Qiyao Zhu[2,5], Swati Jain[1] & Tamar Schlick [1,2,3,4 ✉]

The SARS-CoV-2 frameshifting element (FSE), a highly conserved mRNA region required for correct translation of viral polyproteins, defines an excellent therapeutic target against Covid-19. As discovered by our prior graph-theory analysis with SHAPE experiments, the FSE adopts a heterogeneous, length-dependent conformational landscape consisting of an assumed 3-stem H-type pseudoknot (graph motif 3_6), and two alternative motifs (3_3 and 3_5). Here, for the first time, we build and simulate, by microsecond molecular dynamics, 30 models for all three motifs plus motif-stabilizing mutants at different lengths. Our 3_6 pseudoknot systems, which agree with experimental structures, reveal interconvertible L and linear conformations likely related to ribosomal pausing and frameshifting. The 3_6 mutant inhibits this transformation and could hamper frameshifting. Our 3_3 systems exhibit length-dependent stem interactions that point to a potential transition pathway connecting the three motifs during ribosomal elongation. Together, our observations provide new insights into frameshifting mechanisms and anti-viral strategies.

[1] Department of Chemistry, New York University, 100 Washington Square E, New York 10003 NY, USA. [2] Courant Institute of Mathematical Sciences, New York University, 251 Mercer St, New York 10012 NY, USA. [3] NYU-ECNU Center for Computational Chemistry, NYU Shanghai, 3663 North Zhongshan Road, Shanghai 200062, China. [4] Simons Center for Computational Physical Chemistry, New York University, 24 Waverly Place, New York 10003 NY, USA. [5] These authors contributed equally: Shuting Yan, Qiyao Zhu. ✉ email: schlick@nyu.edu

In less than three years, COVID-19 through its novel infectious agent SARS-CoV-2 has already caused more than 566 million infections and 6 million deaths worldwide. Although the development of multiple vaccines has provided hope for a post-pandemic world, new virus variants with higher infectivity and increased ability to evade the immune system require us to maintain vigilance. Thus, the identification of novel anti-viral therapeutic targets and development of drugs against them remains a priority.

The single stranded SARS-CoV-2 RNA genome of 29,891 nucleotides includes two overlapping and frame shifted open reading frames ORF1a and 1b, which encode for viral polyproteins that begin the viral protein production. To correctly translate both polypeptides, the virus utilizes programmed –1 ribosomal frameshifting (–1 PRF) to stall and backtrack the ribosome by one nucleotide to bypass the stop codon near the start site of ORF1b.

First discovered in the *Rous sarcoma virus* in 1985[1], the –1 PRF stalling of the ribosome is associated with a small (<100-nt) RNA frameshifting element[2]. SARS-CoV-2 similarly employs such a frameshifting element (FSE) located at the ORF1a/1b junction. This FSE consists of a 7-nt slippery site and a downstream 77-nt stimulatory region, which typically folds into an H-type pseudoknot (Fig. 1). The functional importance and high conservation of the FSE make it a promising candidate for anti-viral drugs and gene therapy; for example, in the latest Omicron variant, there are 31 new mutations in the spike gene region with respect to the previous variants, but no changes in the FSE (Supplementary Fig. 1)[3–6]. Whether frameshifting is orchestrated by the FSE acting as a road blocker or through more complex conformational switches remains unknown[7–13]. Hence,

exploring the secondary (2D) and tertiary (3D) structural dynamics of the FSE during translation is essential for both untangling the frameshifting mechanism and developing antiviral strategies.

Unlike the stem-loop structure for HIV-1 FSE[14] or the 2-stem pseudoknot for IBV FSE[15], the assumed structure for SARS-CoV-2 FSE is a 3-stem H-type pseudoknot, where the Stem 1 loop binds the 3′ end to form Stem 2, and Stem 3 lies between them (Fig. 1). This motif has been reported by chemical probing, Cryo-EM, NMR, crystallography[3,16–22], and molecular dynamics (MD)[23–25]. Using our coarse-grained RNA-As-Graphs (RAG) representation as dual graphs[26–29], we assign this pseudoknot motif dual graph 3_6 (Fig. 1)[3,24]. RAG translates double-stranded stems to vertices and single-stranded loops to edges. We use RAG to identify key RNA motifs, design novel RNA motifs from building blocks, and perform inverse folding to transform one RNA motif into another[30–35]. Recent applications of RAG explored the FSE conformational landscape, including RNA mutations to alter the FSE motif[3,24].

Further studies of the SARS-CoV-2 FSE have revealed a complex conformational landscape, with alternative pseudoknots[3,19,36,37] as well as unknotted structures[3,18,19,38–41] (see[3] for a detailed comparison). In particular, our prior modeling and SHAPE chemical reactivity experiments reveal an alternative 3-stem H-type pseudoknot where the Stem 1 loop binds with the 5′ end to form a different Stem 2 (3_3 dual graph), and a 3-way junction where the 5′ and 3′ ends pair (3_5 dual graph)[3]. The three motifs (3_6, 3_3, and 3_5) have common Stems 1 and 3 (though stem lengths vary) but competing Stem 2 (see Fig. 1).

Moreover, our graph and SHAPE studies have emphasized the length dependence of the FSE motifs: for short lengths such

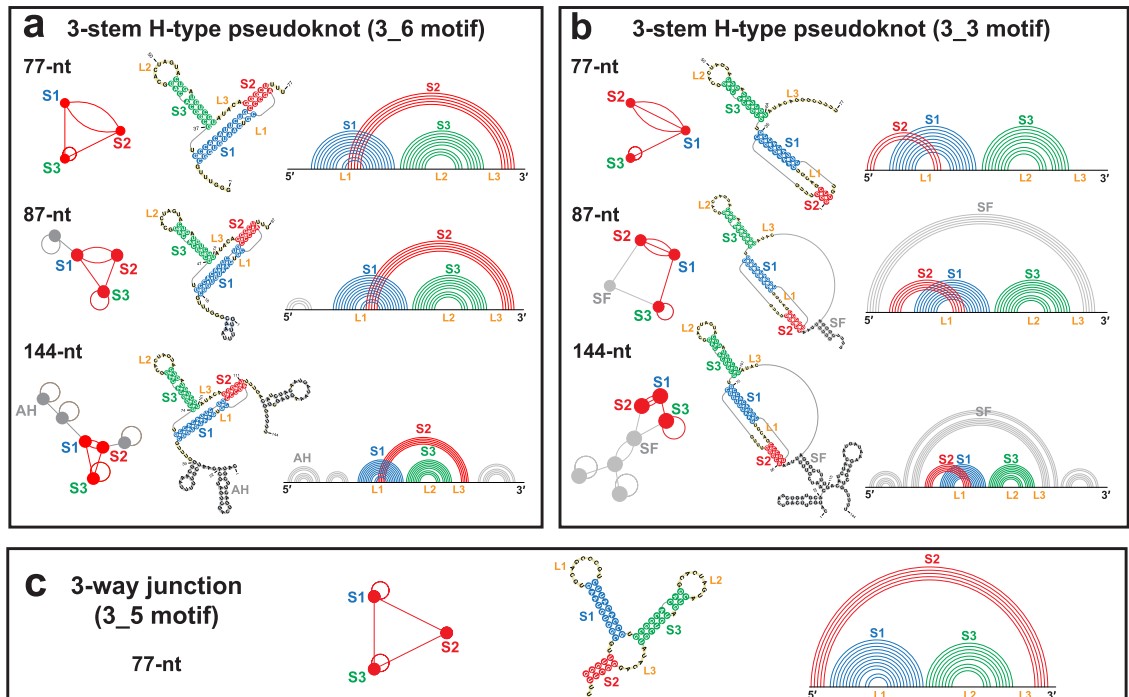

**Fig. 1 Secondary structures of the three FSE motifs. a** 3_6 pseudoknot, (**b**) 3_3 pseudoknot, and (**c**) 3_5 junction at different lengths we study, along with their arc plots and corresponding dual graphs. For 77-nt, the 3_6 pseudoknot, 3_3 pseudoknot, and 3_5 junction have common Stems 1 (blue) and 3 (green), while different Stem 2 (red). The two pseudoknots are classified as H-type[78], where in 3_6 the loop region of Stem 1 binds with the external single-stranded 3′ end, and in 3_3 the Stem 1 loop binds with the 5′ end. For 87-nt, 10 upstream residues are added that include the 7-nt slippery site, and the 3_3 conformation contains an extra flanking stem SF (gray). For 144-nt, 37 upstream and 30 downstream residues are included, and extra stems (gray) are formed, including attenuator hairpin AH for 3_6 and SF for 3_3. Stems are represented as vertices in dual graphs, and loops as edges, with the central 3_6, 3_3, and 3_5 submotifs corresponding to the 77-nt FSE region highlighted in red, and the flanking vertices/edges corresponding to the extra stems/loops in gray.

**Table 1 FSE systems studied in this work. For each motif (3_6, 3_3, and 3_5) and length (77, 87, and 144-nt) combination, we use five 3D prediction programs (R for RNAComposer, S for SimRNA, I for iFoldRNA, V for Vfold3D, and F for Farfar2) to generate starting models.**

| All FSE systems considered (37) and validated (30) | | | | | | |
|---|---|---|---|---|---|---|
| Length | WT 3_6 | WT 3_3 | WT 3_5 | M 3_6 | M 3_3 | M 3_5 |
| 77-nt | *R[1], S[2], *I[3], V[4] | R, S[10], *I[11], V[12] | R[16], S[17], *I[18], *V[19] | R[1'], *S[2'], I[3'], V[4'] | *I[11'] | R[16'], S[17'], *I[18'], V[19'] |
| 87-nt | *R[5], S, *I[6], V | R, S, *I[13], V | N/A | N/A | N/A | N/A |
| 144-nt | R[7], I[8], F[9] | R, I[14], F[15] | N/A | *R[7'], I[8'] | N/A | N/A |

The 19 convergent and validated wildtype MD trajectories and the 11 mutant trajectories are numbered as indicated by the superscript. Representative trajectories for the Results section are marked with asterisks.

as 77-nt without the slippery site, the 3_6 pseudoknot is the dominant motif, and the 3_5 junction is minor; for longer lengths such as 87-nt and 144-nt, conformations containing the 3_6 pseudoknot become minor, while those containing the 3_3 pseudoknot become dominant[3]. We calculated these FSE length/motif landscapes using partition functions of 2D structures predicted with SHAPE reactivity restraints, where we term a particular conformation 3_6, 3_3, or 3_5 according to the central 77-nt FSE fold motif[3]. As in other positive-sense RNA viruses[42–44], structural transitions among these three (and other possible) motifs likely exist and play an important role in frameshifting.

Here we employ several computational structure prediction programs to build candidate FSE 3D models and analyze microsecond MD trajectories of the three motifs at three lengths: 77, 87, and 144-nt (Fig. 1). We consider the 19 wildtype trajectories studied here as pieces of the heterogeneous FSE landscape during ribosomal translation. We also study our motif-strengthening mutants (experimentally validated in[3]) that stabilize each motif over the others (11 trajectories). All starting models and 30 MD trajectories are analyzed with respect to known SHAPE data and available experimental structures.

From the 19 wildtype and 11 mutant FSE trajectories, we identify critical structural features and motions. For the 3_6 pseudoknot, we capture both the L and the linear shapes observed by Cryo-EM and crystallography studies[19,20,22,45], with pseudoknot-stabilizing hydrogen bonds. A threaded ring conformation and a structural switch between the L and linear shapes may play a role in ribosomal pausing and frameshifting. Importantly, we can suppress this transition in our sextuple mutant. From the alternative motifs, especially the 3_3 pseudoknot, we find length-dependent stem interactions that suggest a potential FSE transition pathway during ribosomal translation. All these mechanistic insights help suggest frameshifting mechanisms and open new avenues for anti-viral therapy. Namely, small molecules or gene editing mutations in these key regions could hamper frameshifting: 3_6 threading (3′ helix end of Stem 1), structural switch (Stem 2/3 junction), and pseudoknot-stabilizing interactions (hydrogen-bonded triplets near Stem 2).

## Results

**MD model validation and selection.** Using five 3D prediction programs, we create 26 initial models compatible with our 3 wildtype FSE motifs[3] (3_6, 3_3, and 3_5) at 3 relevant lengths (77, 87, and 144-nt), as listed in Table 1. An initial check for consistent 2D structures with SHAPE data left 23 viable candidates (Supplementary Table 1), which were then subjected to microsecond MD simulations (see "Methods"). MD trajectory convergence and structure validation tests excluded four more cases, with 19 viable trajectories remaining (numbered by superscripts in Table 1). See detailed analyses in SI (Supplementary Figs. 2-8, and Supplementary Tables 2-4).

We consider all these trajectories as parts of the heterogeneous FSE conformational landscape, relevant with length-dependent ribosomal interactions. To simplify our presentation below, we select representative systems (marked with asterisks in Table 1) based on multi-trajectory clustering (Fig. 2) and structure validation (Supplementary Table 3). This results in L-shape trajectories 1 and 5 and linear shape trajectories 3 and 6 for 3_6, trajectories 11 and 13 for 3_3, and compact shape trajectory 18 and elongated shape trajectory 19 for 3_5.

For the motif-strengthening mutants, we only consider trajectories from prediction programs that correspond to our 11 validated wildtype systems, and representative systems are chosen based on Stem 2 lengths (Table 1, see details in Supplementary Tables 5-6 and Supplementary Figs. 9-14).

**Overview of the three fold motifs.** In the following sections, we focus on these representative systems (Table 1). For the 3_6 pseudoknot, we highlight a ring formed by Stem 1 strand and the junctions, and relate the 5′ end threading that may impact ribosomal pausing to the L shape model (Fig. 3). We identify hydrogen-bond networks that stabilize the pseudoknot complex (Fig. 3), and compare our models to the experimental structures (Fig. 4)[19,20,22,45].

For the alternative 3_3 pseudoknot and 3_5 junction, we discuss length-dependent flanking stem or triplet formation (3_3) and the Stem 2/3 interactions (3_5) that provide insights into FSE transitions (Fig. 5).

Inherent motions and motif-strengthening mutants are discussed in Figs. 6 and 7. Notably, a key structural switch between the L and linear shapes for 3_6 that may send frameshifting signals to the ribosome is suppressed in our mutants.

The combined insights suggest target regions for small-molecule binding and CRISPR gene-editing, as well as a transition pathway connecting the three motifs during ribosomal translation (Fig. 8).

**L-shaped and linear 3_6 conformations.** In two recent Cryo-EM structures, the 3_6 pseudoknot exhibits an L shape with a bent Stem 3 from the co-axial plane of Stems 1 and 2 (Fig. 3)[19,20]. Moreover, a ring forms by linking the 3′ strand of Stem 1, and the Stem 1/3 and 2/3 junctions, with the 5′ end threading through the ring hole, which may hamper ribosomal unwinding[19,46].

Here, we observe this threaded L shape in our 77 and 87-nt 3_6 models (Fig. 3, Supplementary Table 4, and Supplementary Fig. 8), stabilized by hydrogen-bond networks. At 77-nt, the Stem 1 loop and the Stem 2/3 junction form a quadruplet to seal the ring top. At 87-nt, another triplet forms between the threaded 5′ end and Stem 1 at the ring bottom. Moreover, the flexible 5′ end folds into a small helix, which is also seen in our SHAPE probing[3] and in the Cryo-EM structure[19].

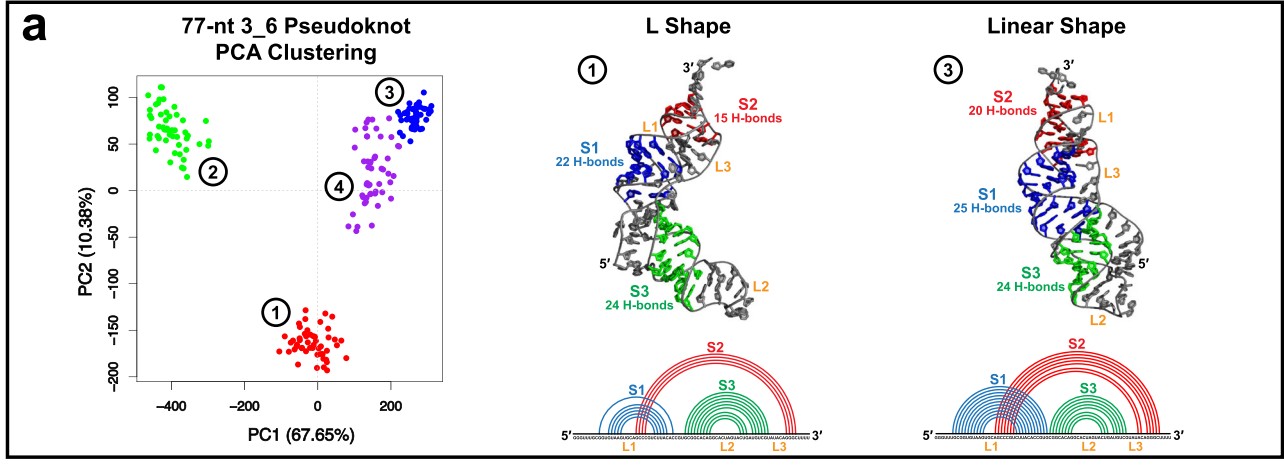

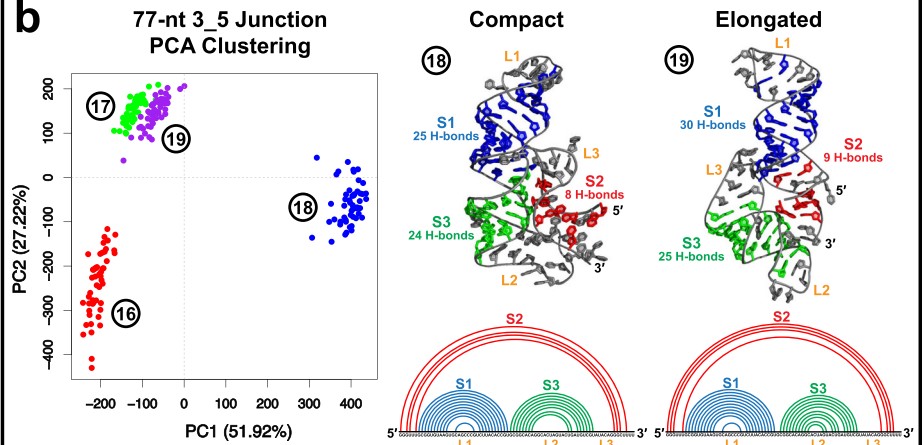

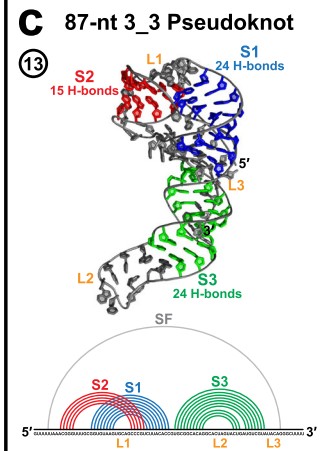

**Fig. 2 Clustering analysis and representative structures for the three FSE motifs.** Multi-trajectory clustering is performed to compare the 77-nt 3_6 trajectories (1–4 in Table 1) and select representative systems, as well as the 3_5 trajectories (16-19 in Table 1), with 50 equally spaced frames extracted from each trajectory. **a** For 3_6, two representative trajectories are chosen, sampling an L and a linear shape, respectively (shown as the largest cluster center structures found by additional single-trajectory clustering, see "Methods"). **b** For 3_5, the two representative models capture compact and elongated shapes. **c** For 87-nt 3_3, only one system maintains the motif throughout the MD simulation. The 2D structures are extracted using 3DNA-DSSR[67], and the numbers of hydrogen bonds formed in the stems (averaged over the last 500 ns of the simulations) are calculated using Gromacs[68].

In contrast to this L shape, we also capture linear shape models observed by crystallography[22,45] at all FSE lengths, where the 3 stems stack vertically (Fig. 3). A similar ring forms, but the hole is narrower and the 5′ end prefers being non-threaded by winding around the structure. Other than ring-stabilizing hydrogen bonds, interactions between the 5′ end and Stem 3 help stiffen the junction to maintain the linear shape.

Although 5′ end threading is also captured in our linear models and the crystal structure (PDB: 7MLX)[22], it is more prevalent in the L shape (Supplementary Table 4). Such threading likely forms before ring closure, as experiments for the 3_6 pseudoknot suggest that Stem 1 forms first and Stem 2 last during FSE folding[46]. Therefore, it is likely that Stem 3 bending (L shape) is associated with threading to avoid steric clashes.

In both the L and the linear shapes, multiple hydrogen bonds act to stabilize the 3_6 pseudoknot complex (Fig. 3). In the 77-nt L shape, the Stem 1/2 and 2/3 junction residues interact to form a short triplex, which is further extended by binding with Stem 2. This triplex stabilizes the loose junctions and links the 3′ end tightly near the Stem 1 loop to maintain the pseudoknot.

In the 77-nt linear shape, Stems 1 and 2 are longer than those in the L shape, but similar triplets are found at the Stem 2 junctions (Fig. 3). These triplets are also seen in the crystal

structures[22,45], indicating their importance in stabilizing the pseudoknot. Another triplet forms between Stem 2 and the 3′ end, anchoring the flexible 3′ end.

**Comparisons to experimental structures**. Comparing our L shape models to the two Cryo-EM structures[19,20], we see global structural similarity in Stems 1 and 2 stacking (Fig. 4). The 88-nt Cryo-EM structure (PDB: 6XRZ, resolution 6.9 Å)[19] has a wider ring than our 87-nt model, possibly due to a more bent Stem 3. Its 5′ end helix is shifted and shorter, and is further away from Stem 3. The 77-nt Cryo-EM mRNA-ribosome complex (PDB: 7O7Z, resolution 5-7 Å)[20] has slightly longer Stems 1 and 3 than our 77-nt model, and its 5′ end is pulled by the ribosome.

The two crystal structures (PDB: 7LYJ and 7MLX, resolution 2.1 Å)[22,45] align well with our 77-nt linear shape model (3-4 Å RMSD). Stem 3 loop is more stretched in the crystal structures, probably because associated residues were mutated to avoid dimerization. Evidence of threading exists in the Roman et al. crystal structure (PDB: 7MLX) but not in the Jones et al. structure (PDB: 7LYJ). Consistent with our comments above, a wider ring hole accompanies the threading to avoid steric clashes.

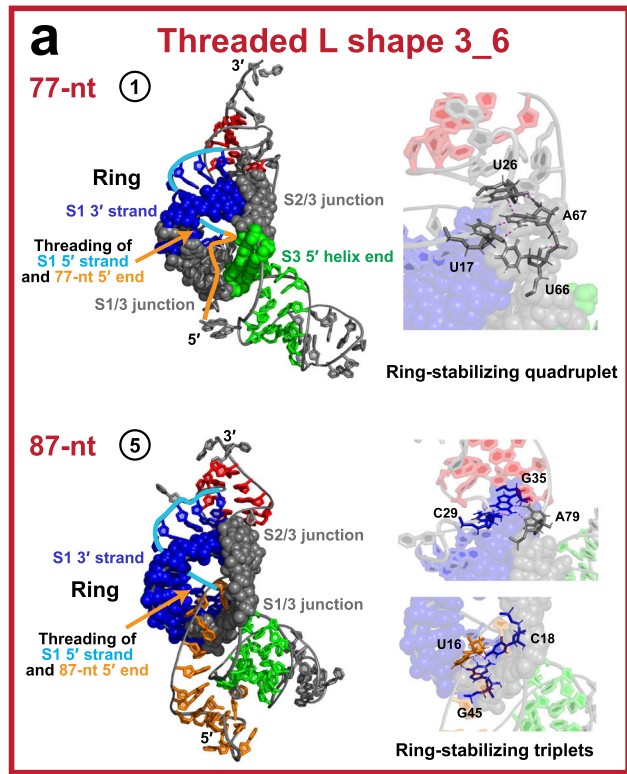

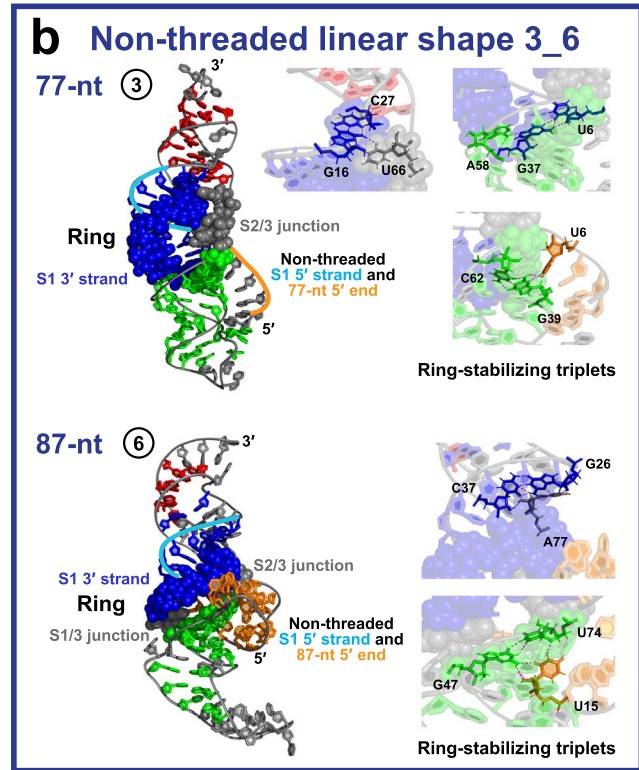

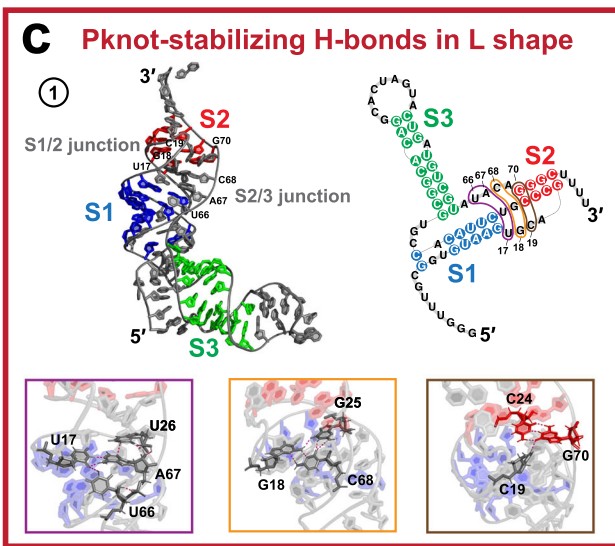

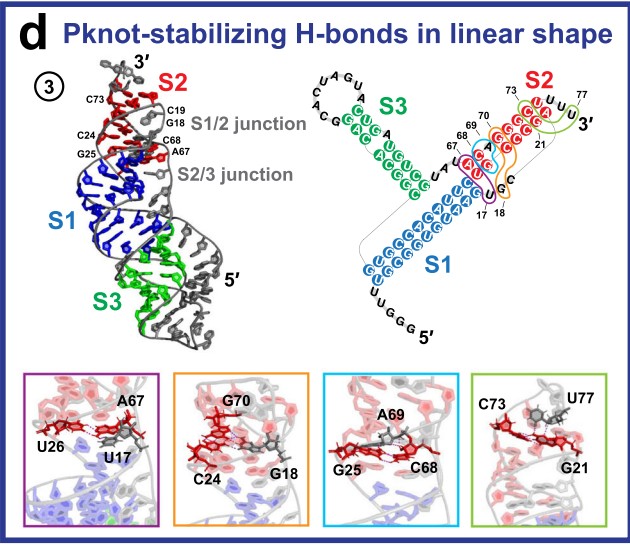

**Fig. 3 Threaded L and non-threaded linear 3_6 pseudoknot ring conformations. a** The threaded 3_6 models have a ring formed by the 3′ strand of Stem 1, and the Stem 1/3 and 2/3 junctions, with the 5′ strand of Stem 1 and the 5′ end threaded through. Ring-stabilizing hydrogen bonds are enlarged. **b** The non-threaded 3_6 models with ring-stabilizing triplets. **c** The 77-nt L shape model has a base quadruplet and two triplets formed at the Stem 1/2 and 2/3 junctions to stabilize the pseudoknot. **d** The 77-nt linear shape model has four pseudoknot-stabilizing triplets formed at the junctions. The systems are numbered according to Table 1.

Overall, our independently developed yet well aligned 3_6 MD structures provide credibility for the following alternative structure modeling.

**FSE transitions suggested by length-dependent 3_3 pseudoknot interactions.** In our SHAPE experiments, the dominant motif in the FSE landscape shifts from 3_6 to 3_3 pseudoknot when the sequence length increases from 77-nt to 87 and 144-nt[3]. This alternative 3_3 pseudoknot contains a different Stem 2 formed by the Stem 1 loop and the 5′ end. At 77-nt, Stem 2 is short with 3 base pairs; at 87 and 144-nt, upstream residues form

2 additional base pairs for Stem 2, and also a flanking stem SF with the 3′ end to further seal the conformation (Fig. 5, more details in Supplementary Figs. 8 and 15).

Consistently, a clear jump occurs for 3_3 Stem 2 hydrogen bond number, when the length increases from 77 to 87-nt, resulting in a stronger Stem 2 of 3_3 than that of 3_6 (Supplementary Fig. 16). A similar trend is observed from the stem interaction energy (Supplementary Fig. 17).

These length-dependent interactions suggest potential motif transitions during ribosomal translation and RNA refolding. Indeed, our 77-nt 3_3 model is identified as an intermediate

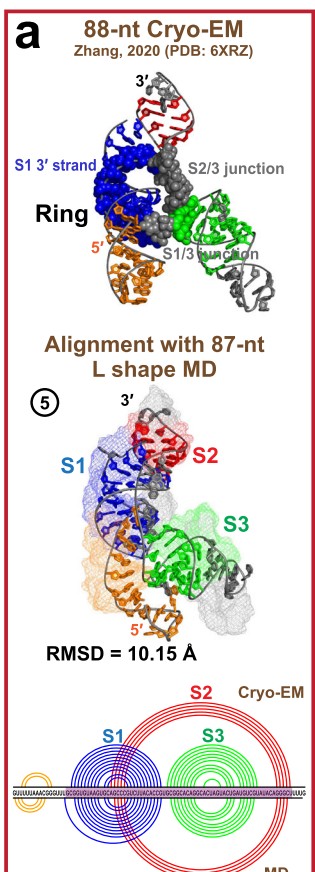
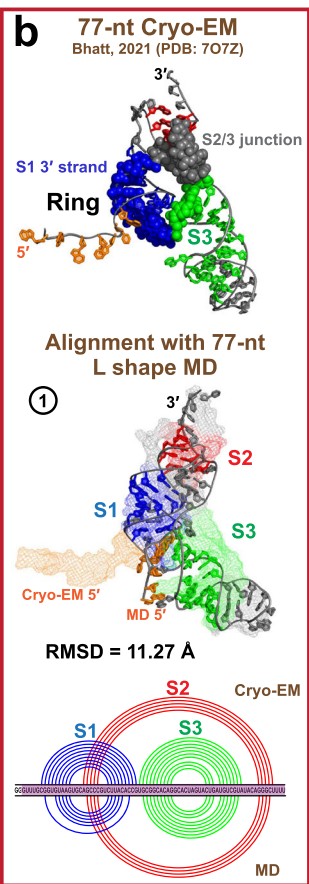
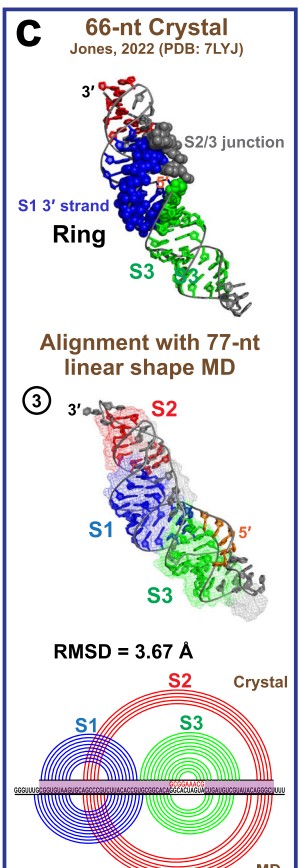
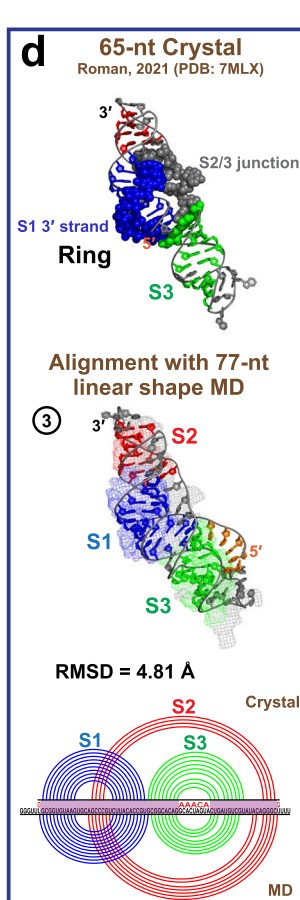

**Fig. 4 MD 3_6 structures compared to four experimental structures. a**, Top The 88-nt Cryo-EM structure (PDB: 6XRZ)[19] in the threaded ring conformation is (Middle) aligned with our 87-nt L shape MD structure (Cryo-EM structure in mesh mode, MD in cartoon), with (Bottom) 2D structural comparisons (Cryo-EM arc plot at top, MD at bottom). The 3D structure alignment is performed by PyMol[77] for the center 67 residues (highlighted in purple in the 2D plot), and the RMSD is computed. **b** Comparison between the 77-nt FSE segment extracted from the mRNA-ribosome Cryo-EM structure (PDB: 7O7Z)[20] and our 77-nt L shape MD structure. **c, d** Comparisons between the crystal structures (PDB: 7LYJ[45] and 7MLX[22]) and our 77-nt linear shape MD structure. The mutated Stem 3 loop residues in the crystal structures are colored red in the 2D structural comparisons at bottom.

structure, in which the 3′ end residues U74 and U75 form two triplets with 3_3 Stem 2 (Fig. 5). In 3_6, the same 3′ end residues pair with Stem 1 loop (A20) to form Stem 2; in 3_5, they pair with the 5′ end (G2 and G1) to form Stem 2. Hence, Stem 2 interactions in all three motifs co-exist in our 77-nt 3_3 model, potentially making this state a starting conformation for a transition from 3_3 to 3_6 or 3_5.

For the 87-nt 3_3 systems, the flanking stem SF formed by the 5′ and 3′ ends blocks Stem 2 of 3_6 and 3_5, and the hydrogen bonding between residue U86 and the Stem 3 base pair C72-G49 keeps the 3′ end away from Stem 2 (Fig. 5). In our 144-nt models, additional stems form to avoid the mixed Stem 2 triplets (Supplementary Figs. 8 and 15). Hence, all these interactions, especially stem SF, must be unwound by the ribosome before the 3′ end is free to form Stem 2 of 3_6 or 3_5.

Together, these insights suggest the following transition pathway during ribosomal elongation: when the ribosome is far away from the FSE 5′ end, the flanking stem SF favors 3_3, but as the ribosome moves to occlude the slippery site, SF is unwound to allow formation of alternative Stem 2, and a transition to 3_6 or 3_5 occurs.

**Elongated and compact 3-way 3_5 shapes.** The 3_5 3-way junction is a minor motif seen in our SHAPE experiments at 77-nt[3], where the 5′ and 3′ ends base pair to form Stem 2. In our MD simulations, we capture both an elongated and a compact 3_5 conformations (Fig. 5).

In the elongated model, Stems 1 and 2 are co-axially stacked, and Stem 3 is longer. A hydrogen-bond network is formed by 5 residues at the 3-way junction to stabilize the helical arrangements, so that Stem 2 would remain around Stem 3 and not interact with Stem 1 loop to form alternatives.

In the compact model, Stems 1 and 3 are stacked, and the Stem 3 loop bends towards the groove of Stem 2. Similar hydrogen bonds at the junction secure the Stem 2 orientation away from Stem 1. These hydrogen bonds must be broken to allow transition to another motif.

**Dominant motions in 3_6: L to linear shape transitions.** Using principal component analysis (PCA), we find that the linear 3_6 models are rather stable, while the L models switch between the two shapes, via bending of Stem 3 (Fig. 6, Supplementary Fig. 18). As the two Cryo-EM structures both capture the L shape and the two crystal structures both exhibit the linear shape, we speculate that the FSE is highly dynamic and may switch between these conformations during frameshifting.

Meanwhile, the pseudoknot complex (Stems 1 and 2) and the threaded ring conformation are maintained throughout this motion, so does the ring-holding triplet at bottom.

Consistent with the above motions, we see a peak in the 3_6 root mean square fluctuations (RMSF) in the Stem 3 loop region for all

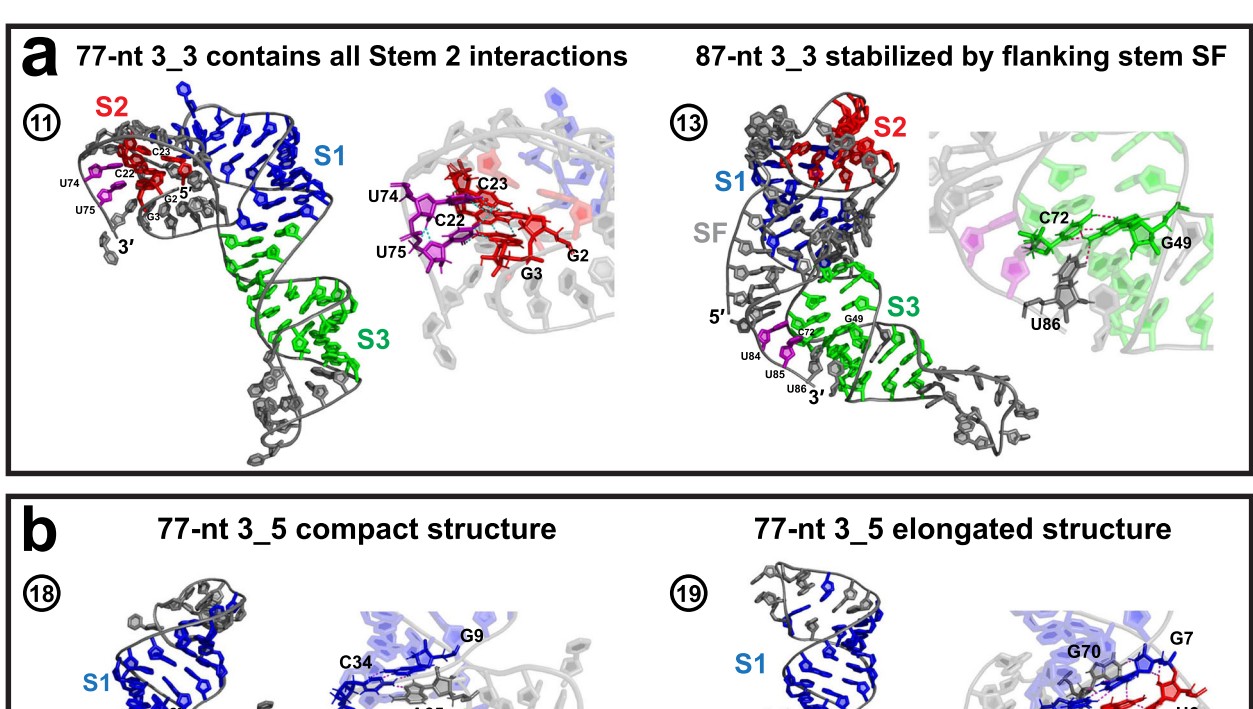

**Fig. 5 Alternative 3_3 and 3_5 motifs. a** The 77 and 87-nt 3_3 pseudoknot structures (model 11 and 13 in Table 1). For 77-nt, two residues in the 3′ end (purple), which are involved in the 3_6 and 3_5 Stem 2, form triplets with the 3_3 Stem 2. For 87-nt, the 5′ and 3′ ends base pair to form flanking stem SF, and the same two 3′ end residues (purple) are locked around Stem 3 by a downstream triplet. **b** The 77-nt 3_5 junction compact and elongated structures (model 18 and 19). The Stem 2 helix formed by the 5′ and 3′ ends is stabilized around Stem 3 by multiple hydrogen bonds.

lengths (Fig. 6). The RMSF, the average number of hydrogen bonds (H-bond), and the interaction energies all indicate that Stem 1 is the strongest, followed by Stem 3, and lastly Stem 2 (Supplementary Figs. 16 and 17).

**Stretching/bending motions in 3_3**. The 3_3 pseudoknot's dominant motion is a combined contraction and stretching caused by the bending of 3′ end and Stem 3 loop (Fig. 6, Supplementary Fig. 18). In this motion, Stems 1 and 2, especially triplets that contain interactions from all three Stem 2 (purple and red residues in Fig. 6), are intact and move in unison. That these triplets are not transient suggests that this may be a stable intermediate during the FSE translation pathway directed by the elongating ribosome.

Comparing to 3_6, we see a higher RMSF peak value in the 3_3 Stem 3 loop region, and more fluctuations in 3_3 Stem 1 region due to the pseudoknot bending.

**Stem 3 twisting in 3_5**. For the 3_5 junction, both the elongated and the compact models experience bending of Stem 1 loop and twisting of Stem 3 (Fig. 6, Supplementary Fig. 18). As a result, the structure becomes more compact, and Stem 2 is closer to Stem 3. All the hydrogen bonds that lock the Stem 2 orientation (Fig. 5) are maintained, so that a stable 3_5 motif remains. Peak RMSF in the loop regions, and low values in the 5′ and 3′ ends are notable.

Overall, all three conformations have stable Stem 1, flexible Stem 3 loop, and relatively stable Stem 2 regions. The triplets and hydrogen bonds are mostly maintained throughout the simulations, and this helps stabilize key features such as the ring of 3_6 and the combined Stem 2 interactions in 3_3.

**Minimal mutations to stabilize the 3_6 linear shape**. Our predicted mutations confirmed by SHAPE probing in our previous study were designed to suppress conformational transitions and stabilize specific motifs over all alternatives, for the 77 and 144-nt 3_6 pseudoknot, 77-nt 3_3 pseudoknot, and 77-nt 3_5 junction (Table 1)[3,24]. Our dynamics analyses below of these mutants compared to the wildtype trajectories help interrogate the mechanisms and consequences of structural stability.

The 6 mutations in the 77-nt 3_6 pseudoknot-strengthening mutant (PSM) include 4 mutations ([G18A, C19A, C68A, A69C]) that lengthen Stem 2 by up to 4 base pairs (Table 2) and 2 mutations at the 5′ end to exclude alternative 3_3 and 3_5 Stem 2. Comparing the mutant with longest Stem 2 (9 base pairs) to its corresponding wildtype model, we observe a dramatic transformation from L shape (wildtype) to linear shape (Fig. 7). Indeed, all 3_6 mutant systems adopt this linear shape (Supplementary Table 6, Supplementary Fig. 14), and the structural switch between the two shapes has been suppressed in most systems (Supplementary Fig. 19).

For the 144-nt 3_6 PSM, one additional mutation in the downstream region suppresses formation of competing stems[3]. The

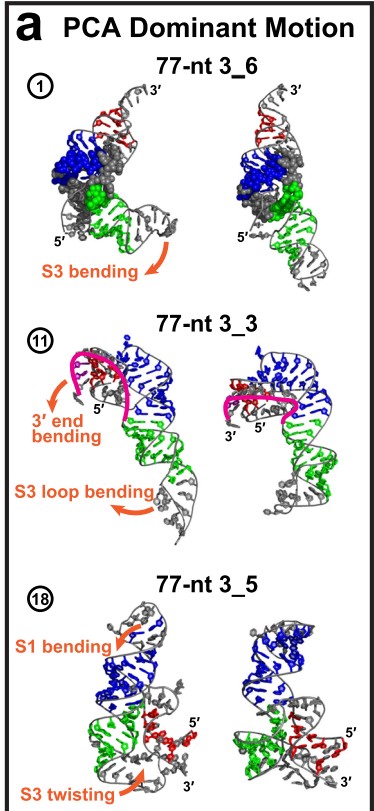 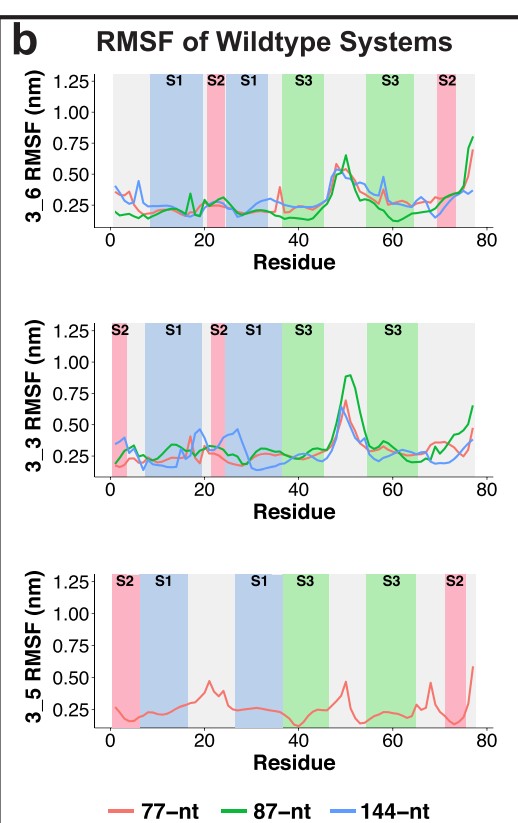

**Fig. 6 Dynamic analysis of the wildtype 3_6, 3_3, and 3_5 systems. a** Dominant motions of the threaded L shape 77-nt 3_6 pseudoknot, 77-nt 3_3 pseudoknot, and 77-nt compact 3_5 junction extracted by principal component analysis (PCA). **b** Flexibility of the three conformations as reflected by root mean square fluctuations (RMSF). For the 3_6 and 3_3 pseudoknots, the RMSF is shown for the common 77-nt region at various lengths; for 3_5 junction, RMSF at 77-nt. The different stem regions are colored and labeled.

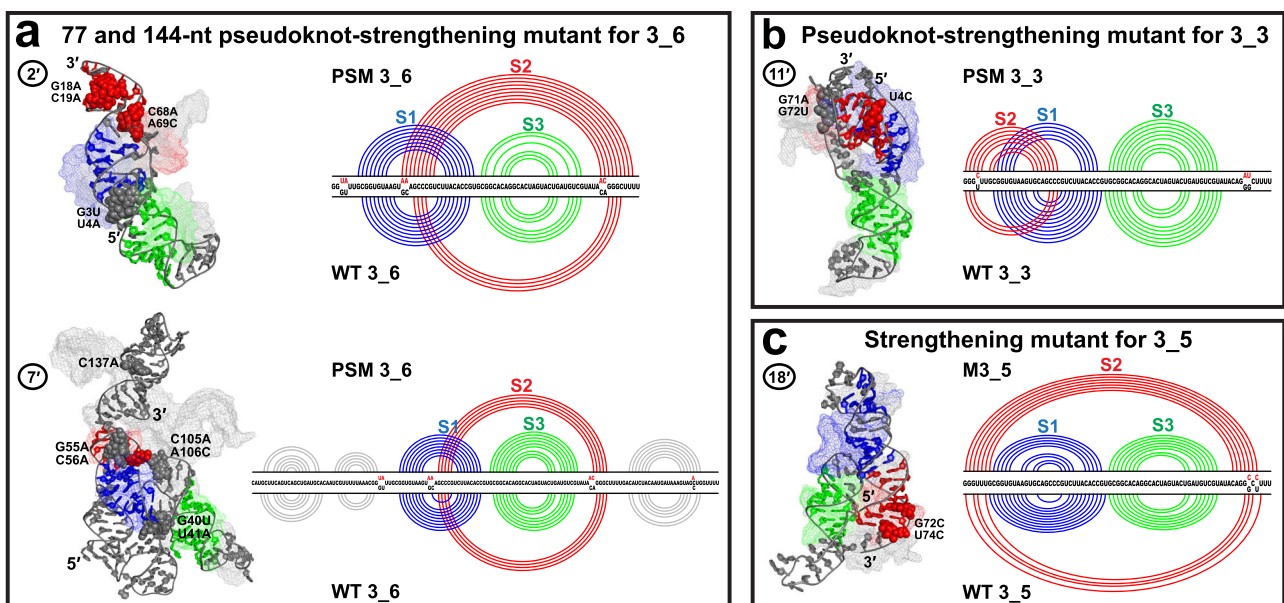

**Fig. 7 Comparison of the motif-strengthening mutants with the wildtype systems. a** For the 3_6 pseudoknot, both the 77-nt and 144-nt PSM are shown in cartoon mode with their wildtype systems aligned in mesh (by PyMol[77] for the 77-nt region). The mutations are highlighted as spheres in pseudoknot-strengthening mutant (PSM) structure and labeled. The 2D structure comparison is also provided with PSM at top and wildtype at bottom. Comparisons for the (**b**) 77-nt 3_3 PSM and (**c**) 77-nt 3_5 mutant are shown in similar manner. Systems are numbered according to Table 1.

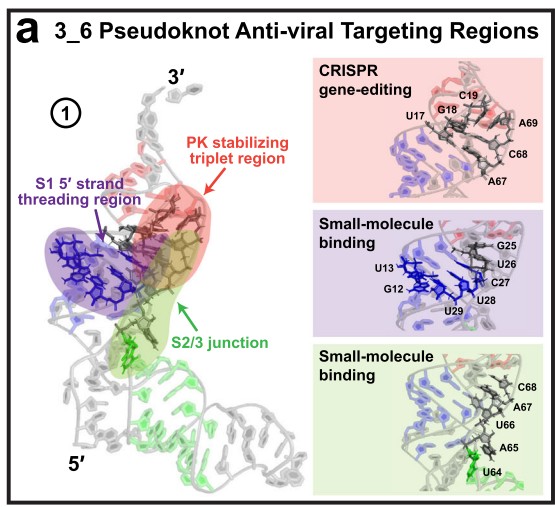

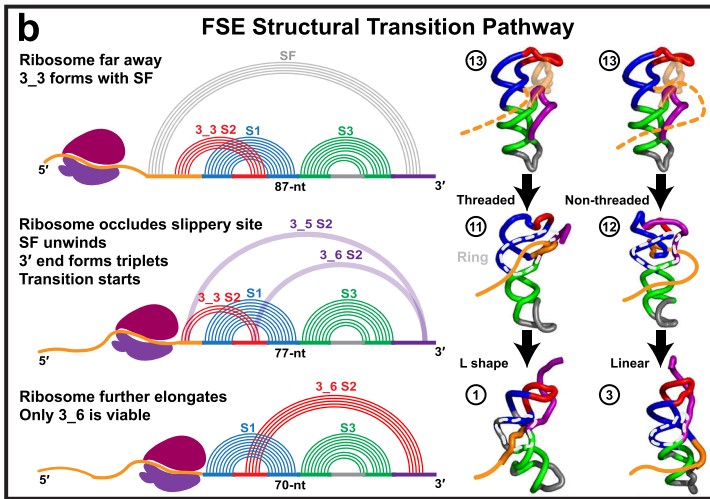

**Fig. 8 Implications of the unraveled structures and motions to anti-viral therapeutics and frameshifting mechanisms. a** Anti-viral target regions in the 3_6 pseudoknot. **b** Proposed structural transition pathway for the SARS-CoV-2 frameshifting element. The loop region of Stem 1 is colored red, the 5′ end orange, and the 3′ end purple. The 87-nt 3_3 model shown at top has the two ends base paired to form stem SF, and a potential threaded and non-threaded 5′ end is plotted as dashes. The 77-nt 3_3 models in the middle have triplets that contain alternative Stem 2 interactions, and the ring starts to close (circled in gray), with the 5′ end threading through or winding around. The 77-nt 3_6 models at bottom has a threaded L shape and a non-threaded linear shape. The 5′ ends in these models are extended just for illustration. The trajectory numbers of the representative conformations are indicated as defined in Table 1. The 3D models are drawn with PyMol using the cartoon rendering option[77].

| Table 2 Comparison of the motif-strengthening mutants and the wildtype systems. | | | | |
|---|---|---|---|---|
| **Trajectory** | **WT S2** | **Mutant S2** | **Base pairs involving mutations** | **Structural features** |
| 77-nt 3_6 PSM [G3U, U4A, G18A, C19A, C68A, A69C] | | | | |
| 1′ | 4 | 4 | — | Threaded, linear |
| 2′ | 4 | 9 | A18-U76, A19-U75, G25-C69, U26-A68 | Threaded, linear |
| 3′ | 7 | 7 | A19-U75, G25-C69 | Non-threaded, linear |
| 4′ | 4 | 8 | A18-U76, A19-U75, G25-C69, U26-A68 | Non-threaded, linear |
| 144-nt 3_6 PSM [G40U, U41A, G55A, C56A, C105A, A106C, C137A] | | | | |
| 7′ | 4 | 5 | G62-C106 | Non-threaded, linear |
| 8′ | 5 | 4 | A56-U112 | Non-threaded, linear |
| 77-nt 3_3 PSM [U4C, G71A, G72U] | | | | |
| 11′ | 3 | 7 | C4-G21 | No Stem 2 triplets |
| 77-nt 3_5 Mutant [G72C, U74C] | | | | |
| 16′ | 3 | 7 | G1-C74, G3-C72 | Elongated |
| 17′ | 3 | 7 | G1-C74, G3-C72 | T shape |
| 18′ | 4 | 7 | G1-C74, G3-C72 | Elongated |
| 19′ | 3 | 6 | G3-C72 | Elongated |

For each mutant, the mutations, the trajectories as numbered in Table 1, the wildtype and mutant Stem 2 lengths, the newly formed Stem 2 base pairs involving the mutated residues, and the structural features are listed.

central 3_6 pseudoknot region aligns well between the wildtype and mutant systems, both adopting the linear shape (Fig. 7). The major difference occurs in the upstream region: in the wildtype, upstream and downstream stems form on the same side of the central 3_6 pseudoknot; in the mutant, they are on different sides, due to our [G40U, U41A] mutations. From PCA, we see a relatively stable central 3_6 pseudoknot, while quite flexible upstream and downstream stems in the mutant (Supplementary Fig. 19).

As both our 77 and 144-nt 3_6 mutants adopt linear conformations, we hypothesize that this may be a more stable conformation, by separating the 5′ and 3′ ends further away from each other to avoid alternative 3_3 and 3_5 Stem 2.

**3_3 mutant to eliminate alternative Stem 2 interactions**. In our 77-nt 3_3 PSM, a large increase of Stem 2 length from 3 to 7 base pairs is induced by mere three mutations [U4C, G71A, G72U] (Table 2, Supplementary Fig. 14). The first mutation enhances the

3_3 Stem 2 and the others avoid alternative 3_6 and 3_5 motifs. The main structural changes are a vertical 5′ end between the Stem 1 loop and helix instead of staying horizontal below, compact Stems 1 and 2, and elimination of triplets formed by the 3′ end with Stem 2 (Fig. 7). Hence, our mutations stabilize the 3_3 motif. The dominant motion occurs in the Stem 3 region (Supplementary Fig. 19).

**Elongated 3_5 mutant**. Our 77-nt 3_5 mutant with only 2 mutations [G72C, U74C] also enjoys a considerable enhancement of Stem 2 from 3-4 base pairs to 6-7 (Table 2). Though Stem 2 remains close to Stem 3, the loop region of Stem 3 now stretches out, leading to an elongated conformation (Fig. 7). Moreover, the co-axial stacking changes from Stems 1 and 3 in the wildtype to Stems 1 and 2 in the mutant. Indeed, all four 3_5 mutant models adopt a stacking of Stems 1 and 2, and all resemble the elongated

conformation except one (Supplementary Fig. 14). The dominant motion is bending of Stem 1 and 3 loops (Supplementary Fig. 19).

Overall, our enhanced Stem 2 in the three mutants leads to dramatic structural changes, especially for the 77-nt 3_6 and 3_5 systems. PCA analysis reveals stabilization of the linear shape in 3_6 PSM. For the 77-nt 3_3 mutant, triplets associated with possible structural transitions are eliminated.

## Discussion

From our 30 molecular dynamics trajectory ensemble of SARS-CoV-2 FSE systems, we can begin to piece together aspects of the complex conformational landscape. In particular, our simulations extend beyond 3D structure models for the prevalent 3_6 pseudoknot in the literature[3,16–21,23,24,37,47], by providing the first viable 3D models for the alternative 3_3 pseudoknot and 3_5 junction, as well as the motif-strengthening mutants[3]. Moreover, guided by our prior SHAPE probing, our analyses of the 3 FSE motifs at different lengths help examine the conformational changes that might occur during ribosomal translation.

Our 3_6 models exhibit two distinct conformations: a threaded L shape that resembles the Cryo-EM structures[19,20], and a non-threaded linear shape that aligns well with the crystal structures[22,45] (Figs. 3 and 4). An interconversion between the L and linear shapes can be achieved by bending of Stem 3, as revealed by our PCA motion analysis (Fig. 6). Moreover, our mutants stabilize the linear shape and suppress the interconversion (Fig. 7). Importantly, our 3_3 models show length-dependent stem interactions: an intermediate structure containing base triplets from all three alternate Stem 2 at 77-nt, while a classic 3_3 stabilized by flanking stem SF at 87 and 144-nt (Fig. 5). Our 3_3 triple mutant successfully lengthens Stem 2 and eliminates these triplets.

The combined insights suggest the following FSE structural transition pathway relevant to ribosomal translation (Fig. 8). When the ribosome is far away from the FSE region, the dominant conformation is a 3_3 with stem SF. As the ribosome approaches and occludes the slippery site, stem SF is unwound, and the 3′ end moves to the 3_3 Stem 2 region to form triplets, initiating the structural transition to 3_6 or 3_5. In this step, the ring starts to close at the Stem 1/3 and 2/3 junctions, and the 5′ end can either thread through the ring hole or wind around Stem 3. When the ribosome further elongates, the 5′ end becomes completely occluded, and only 3_6 remains viable. If the 5′ end is threaded, to avoid steric clashes, Stem 3 would bend to widen the ring hole; if non-threaded, the 5′ end would interact with Stem 3 to stiffen the junction and hold the linear shape (Fig. 3).

Such length-varying considerations of RNA are relevant to ribosomal translation and co-transcription in general, where the RNA can fold into different transient structures to accomplish various functions[48,49]. For SARS-CoV-2, this FSE transition pathway may be associated with regulatory functions. The timescale at which the transitions occur depends on the scale of conformational rearrangements. Base pairing or tertiary structure changes occur on microsecond to second range. Major interconversions between secondary structures occur on millisecond and longer[50]. Given that the ribosome pauses ~2.8s between translocations[51], this time allows for the structural switches and transitions discussed here to occur. Enhanced sampling simulations are required to probe such transitions.

The biophysical insights from our work also suggest three general therapeutic approaches using small-molecule binding and CRISPR gene-editing (Fig. 8). The first anti-viral strategy is to alter the 3_6 pseudoknot plasticity. By mutating residues that form the pseudoknot-stabilizing hydrogen bonds (Stem 1/2 and 2/3 junctions, Fig. 3), we can further strengthen or destroy the

pseudoknot. Since conformational plasticity has a large impact on frameshifting efficiency[9], this should interrupt the frameshifting process. Indeed, Bhatt et al. achieve a significant reduction in frameshifting efficiency by mutating these junctions[20]. In our prior SHAPE probing, mutations in this region modify the conformational landscape to 100% 3_6[3]. Both studies underscore the sensitivity of the 3_6 pseudoknot and its associated frameshifting to these junctions, which define good targets for CRISPR gene-editing.

The second approach is to strengthen the 5′ end threading in the 3_6 ring conformation. A higher unfolding force is required when threading exists[46], so strengthening the threading may increase the mechanical barrier for translation. Recently, two *alkaloids* (*emetine* and *cephaeline*) predicted to bind the threading initiation site were found to inhibit SARS-CoV-2 viral replication[52]. Hence, the 3′ helix end of Stem 1, which we find to seal the ring and initiate threading, defines a target binding region to impede ribosomal translation (Fig. 8).

The third approach is to target the 3_6 pseudoknot structural switch between the L and linear shapes. In the mRNA-ribosome Cryo-EM structure captured during translation[20], the L shape 3_6 wedges at the mRNA entry channel and resists unwinding by the helicase, which generates tension on the upstream mRNA[20]. This structural switch might then enhance fluctuations of this tension and send frameshifting signals to the ribosome. When switching from the L to linear shape, residues in the Stem 2/3 junction are exposed (Fig. 6); small molecules like *MTDB*[10,53] can thus block the switch and hamper frameshifting. Another option is to deploy our 3_6 mutant, which assures a stabilized linear shape (Fig. 7).

In sum, by analyzing the hydrogen bonding interactions and motions of different 3_6 systems, we offer three strategic anti-viral targeting regions: the 3′ helix end of Stem 1 and Stem 1/2 and 2/3 junction residues (Fig. 8). Although several drugs/small molecules have been shown to inhibit SARS-CoV-2 frameshifting, including *MTDB*[17,53,54], *alkaloids*[52], *Merafloxacin*[55], *Ivacaftor*, and *Huperzine A*[56], they are mainly found by high-throughput drug screening, so the underlying inhibition mechanism is unexplained and, in some cases, the binding regions are unknown. Our targeting regions above emerged from mechanistic considerations.

Of course, like every computational approach, there are inherent limitations and approximations in the modeling and simulations performed here. The SHAPE experiments used to deduce the 3 conformers cannot directly reveal the 2D structure, but only provide nucleotide reactivities as restraints for guiding the 2D structure predictions. Our MD trajectories are started from predicted 3D models, but each model is anchored to SHAPE, Cryo-EM, and/or crystallographic data, rendering the results credible (see Fig. 4 and SI). Though microsecond simulations are near the state-of-the-art, each trajectory provides only a local sampling of the complex, multidimensional thermodynamic space. Enhanced sampling simulations could be used to further unravel the complex rugged landscape of the FSE. While we used simple ionic environments to avoid biases, a more extensive investigation with other ions like $Mg^{2+}$ and $K^+$ is warranted. Nevertheless, taken together, the 30 trajectories here contribute to an emerging view of the FSE conformational landscape as the ribosome elongates.

Our methods and analyses also extend to other viruses and diseases more broadly. mRNA-based therapeutics have already demonstrated success in vaccines and drugs to treat viral diseases and cancers, with the advantage of fast production and flexible design[57–59]. Highly-conserved functional RNAs like the frameshifting element define good drug targets. With our continuously evolving computational and experimental toolkit for investigating RNA systems at increasing complexity, biophysical approaches will continue to contribute to disease diagnostics and treatment.

## Methods

**RAG notation and mutations.** In our RNA-As-Graphs (RAG) framework, RNA secondary structures containing pseudoknots are represented as dual graphs[26]. Each stem (≥2 base pairs) denotes a vertex, and every single strand or loop is an edge (hairpins are self-loops; 1-nt bulges, internal loops with two 1-nt strands, and dangling ends are ignored). Every non-isomorphic dual graph is assigned an identifier V_n, where V is the vertex number and n is a unique motif identifier. Our dual graph library consists of over 100,000 unique dual graphs for 2-9 vertices[29].

To design RNAs with minimal mutations that make the FSE fold in silico onto a target dual graph, we developed our inverse folding program for dual graphs Dual-RAG-IF[24,35]. For manually selected mutation regions and a target 2D structure, Dual-RAG-IF uses a genetic algorithm to generate a pool of candidate RNA sequences with mutations. These candidates are screened by 2D prediction programs to ensure the correct graph folding, and are optimized for minimal mutations. Detailed design of the mutants is described in[3,24].

**FSE lengths and conformations.** We model the FSE structure at three sequence lengths: 77-nt without the 7-nt slippery site, 87-nt with the slippery site plus 3 additional residues at the 5′ end, and 144-nt with the slippery site plus 30 additional residues at each end. We perform MD simulations for all three conformations for the 77-nt FSE. (Even though the 3_3 pseudoknot was not observed at this length, we study it for comparison with other lengths.) For 87 and 144-nt, we model the 3_6 and the 3_3 conformations, with additional stems formed by the upstream and downstream nucleotides (Fig. 1).

Besides wildtype FSEs, we also model four motif-strengthening mutants predicted previously[3,24]: 77-nt 3_6 PSM with 6 mutations [G3U, U4A, G18A, C19A, C68A, A69C], 144-nt 3_6 PSM with an additional mutation C137A, 77-nt 3_3 PSM with 3 mutations [U4C, G71A, G72U], and 77-nt 3_5 mutant with 2 mutations [G72C, U74C].

**2D and 3D FSE structures.** The 2D structure of the wildtype 77-nt 3_6 pseudoknot is predicted by PKNOTS[60], and all other 2D conformations are modeled by ShapeKnots from RNAstructure package with SHAPE reactivities incorporated[3,61].

Corresponding 3D structures are predicted, with the sequences and the 2D structures as input using RNAComposer[62], Vfold3D[63], SimRNA[64], and iFoldRNA[65] for 77 and 87-nt, and RNAComposer, iFoldRNA, and Farfar2[66] for 144-nt, as SimRNA and Vfold3D failed to produce models for this length (see Table 1). For 3D structure prediction programs that give multiple structures as output, the first structure that retained the correct motif is selected as the initial 3D model.

**Initial 3D model validation.** We extract 2D structures of the initial predicted models using 3DNA-DSSR[67]. If the central 77-nt FSE region does not fold into the correct motif (3_6, 3_3, or 3_5), the model is rejected. We also calculate the Hamming distance between each model's 2D structure and the input (SHAPE) 2D structure. Models with Hamming distances >10 are rejected.

**Molecular dynamics details.** We use Gromacs 2020.3 and 2020.4[68], with the Amber OL3 forcefield[69]. The systems are solvated in the cubic box with TIP3P water model, with a buffer of 10 Å from the RNA molecule[70]. The systems are first neutralized with sodium ions and set to a 0.1M NaCl bulk concentration with additional Na+ and Cl− ions. The systems are energy minimized via steepest descent and equilibrated under NVT (300 K) and NPT (1 bar and 300 K) ensembles for 100 ps each. Simulations are run with a timestep of 2 fs and a SHAKE-like LINCS algorithm[71] with constraints on all atom bonds. The Particle Mesh Ewald method[72] is used to treat long-range electrostatics. Production runs are performed for 1 ~ 1.5 $\mu$s under NPT to ensure stable RMSD. Structures from the last 500 ns of each simulation are used for analysis.

Clustering is performed on frames every 200 ps for RNA non-H backbone atoms, using the Gromos clustering method with 2, 2.5, 3, and 3.5 Å cutoffs. The largest cluster center structures (cutoff of 2.5 Å for 77-nt and 87-nt systems or 3.5 Å for 144-nt systems) are extracted from MD simulations to show and analyze in Results and Supplementary Information. The cutoffs are chosen to ensure that all simulations for the same dual graph topology produce a feasible number of clusters with outlier structures excluded. See Supplementary Fig. 7 for more details.

PCA motion analysis (on structures every 250 ps), whole system density, calculations of Rg, RMSF, interaction energy (sum of short-term Lennard-Jones and Coulomb interactions) between the two strands within each stem, and the number of hydrogen bonds in each stem are performed via Gromacs 2020.3[68]. The 2D structures are extracted using 3DNA-DSSR[67]. To compare different MD trajectories of 77-nt 3_6 or 3_5, multi-trajectory clustering is conducted using structures extracted from the last 500 ns simulations by R package *bio3d*[73].

All microsecond MD simulations were conducted on the Prince or Greene supercomputer clusters at the New York University High Performance Computing facilities. Each compute node in the Prince cluster is equipped with two Intel Xeon E5-2690v4 2.6 GHz CPUs (Broadwell, 14 cores/socket, 28 cores/node) and 125 GB memory. Each simulation is performed with seven to eight dedicated nodes (i.e., 196-

224 cores), so the simulations complete in 7-10 days. Each compute node in the Greene clusters is equipped with two Intel Xeon Platinum 8268 24C 205W 2.9 GHz CPUs with 48 cores/node and 192 GB memory. Each simulation is performed with 30 nodes using 32 cores each, so that the simulations complete in 2-4 days.

**MD trajectory validation.** To monitor convergence of the MD trajectories, the system density is examined for NPT ensemble. Next, all simulations are examined for a steady state RMSD[74] plateau, and those with large RMSD fluctuations are extended until reaching a steady state over the last 500 ns. To further evaluate structural convergence, time series of Rg and eRMSD are calculated via Gromacs[68] and Barnaba[75], respectively. The evolution of hydrogen bonds over MD simulations is also used to assess structural stability every 100 ns. The cumulative number of hydrogen bonds is plotted against residue number. It adopts a mountain-like shape, where the number increases as new hydrogen bonds form in the 5′ strand of the stem and decreases in the 3′ strand.

All stable MD trajectories are further validated for 2D and 3D structures. For wildtype systems, we monitor graph motifs, 2D structures, and 3D clashes. Clashscores are calculated as the number of serious steric clashes per 1000 atoms using MolProbity[76]. Models having wrong motifs are rejected, and those having Hamming distances >10 or clashscores >5 are tagged. In addition, all the 3_6 models are aligned to the available 4 experimental structures using PyMol[77]*align*, with RMSD calculated. To validate the mutant systems, we check both the graph motifs and the 2D structures.

**Reporting summary.** Further information on research design is available in the Nature Research Reporting Summary linked to this article.

## Data availability

Complete data are present in the paper and/or associated Supplementary Materials. Additional information including initial 3D prediction models and validated trajectory cluster centers are shared in Zenodo (https://doi.org/10.5281/zenodo.6625172).

Experimental RNA structures analysed in this paper are available in the Protein Data Bank under accession codes 6XRZ, 7O7Z, 7LYJ, and 7MLX.

## Code availability

The codes used to discover motif-strengthening mutations are available in the GitHub Schlicklab repository (https://github.com/Schlicklab/Dual-RAG-IF).

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

## Acknowledgements

We thank Shereef Elmetwaly for technical assistance and David Ackerman, Stratos Efstathiadis, and Shenglong Wang from the NYU High-Performance Computing facilities for providing our group dedicated resources to perform this work.

We gratefully acknowledge funding from the National Science Foundation RAPID Award 2030377 from the Division of Mathematical Science and the Division of Chemistry, National Science Foundation Award DMS-2151777 from the Division of Mathematical Sciences, National Institutes of Health R35GM122562 Award from the National Institute of General Medical Sciences, and Philip-Morris International to T. Schlick.

## Author contributions

T.S. conceived the project and supervised the study. Q.Z. predicted FSE 2D structures, S.J. predicted initial 3D models, S.J. and S.Y. performed molecular dynamics simulations, and S.Y. and Q.Z. validated 3D models and MD trajectories. All authors analyzed the data and wrote the manuscript, and Q.Z. and S.Y. prepared the figures.

## Competing interests

The authors declare no competing interests.
