## [Peer Review File · Nature Communications]

Length-dependent motions of SARS-CoV-2 frameshifting RNA pseudoknot and alternative conformations suggest avenues for frameshifting suppressionEditorial Note: Parts of this Peer Review File have been redacted as indicated to remove third-party material where no permission to publish could be obtained.

REVIEWER COMMENTS

Reviewer #1 (Remarks to the Author):

The manuscript treats the novel idea that conserved SARS-CoV-2 RNA regions may be good drug targets, and in particular the frameshifting element (FSE).

The authors derive three frameshift conformations and perform MD, leading to the conclusion that ring-like conformations may exist. The work is quite computationally thorough and may form a basis for the design of inhibitors. I recommend publication subject to the following aspects:

My main questions are as follows:

- (a) There appears to be no experimental conformation of the alternative conformations found. Some discussion of how this could be achieved would be merited.
- (b) Is there any evidence for convergence of the MD with regards to the conformation space of the FSE.
- (c) Can the authors say something about the transition path between the conformations. PCA is a start for this but typically does not go all the way.

Editing for English quality would be useful.

Reviewer #2 (Remarks to the Author):

The manuscript by Schlick and co-authors presents the study of several putative structures of the SARS-CoV-2 frameshifting pseudoknot using graph theory and molecular dynamics to shed light on possible dynamic transition of this element and its role in translation of the viral genome.

This element is an important therapeutical target since it regulates the expression of half of the viral proteins, and it is well conserved in all the known variants of the virus as well as in related viruses such as SARS-CoV. Moreover, the mechanism of frameshift is common to several other viruses and its mechanism is still largely unknown.

Using graph theory to represent RNA secondary structures including pseudoknots the authors are able to identify three possible architectures of the 77-nt pseudoknot and of two larger systems containing the frameshifting sequence and additional portions of the sequences elongating the element.

The approach is interesting as the comparison of elements of different length can bring light to the dynamic transformations that the element undergoes as it is read by the ribosome, and therefore made shorter.

I have however important reservations on the results obtained mainly due to the choice of the initial configurations. The authors use several folding predictions software to build possible 3D structures for the three architectures found by graph theory. While the 2D structures are supported by chemical probing data, giving them some certainty on their actual existence, I could not find in the manuscript nor in the SI some convincing evidence that the predictions made for the 3D are plausible. Indeed, the different prediction tools propose significantly different structures for the various systems. The authors seem to use all these structures in their analysis (which in my view is a good thing), but in my opinion given the uncertainties in these initial structures the results should be presented with more caution and making more connection/comparisons with existing experimental data.

Probably because of the large number of initial structures used, I often found myself lost in the result sections not knowing what was used when and what was a result of MD simulations. In its present

state I think the result section is hard to follow, at least up to the discussion on mutations where things become clearer.

My main suggestions would therefore be to clearly substantiate the initial structures giving the experimental evidence of their relevance and to revise the result section accordingly and also to make it easier to follow.

A few remarks in particular:

1. Can the authors compare the predicted initial structures against the SHAPE data used also in the 2D predictions? It is true that these data are already included in the prediction, but they have undergone several steps and approximations in making the 2D prediction. I think it would be worth to compare the secondary structures of the predicted 3D structures (both initial and after MD) to verify that the pairings predicted by SHAPE are actually there, especially for the possible 3D contacts which are harder to include in secondary structure predictions. I think that such an analysis would greatly strengthen the validity of the results. Possibly using also other kind of experimental data, it will then be useful to clearly state among the possible initial structures which are the ones supported by experimental evidence (and what) and which are purely putative.

2. Always on the choice of the initial structures, not all prediction tools have the same level of detail as some of them are coarse grained. However, the proposed structures are used assuming the same validity for all predictions, which is debatable. For the models coming from coarse graining an additional validation of the relevance of the proposed structure should be presented.

3. Part of the discussion focuses on the threading of a single strand through a ring formed by the pseudoknot. Again, I think that the existence of this threading should be substantiated with more experimental evidence. If clear evidence for this configuration exists it would be useful to spend a few words detailing it.

4. When presenting the three possible architectures, the authors discuss which one is dominant over the others depending on the length of the sequence. However, I could not find how this dominance is assessed/calculated. Does it have to do with the partition function in the 2D predictions? This point should be clarified.

5. The choice of the three representative models on page 3, 139-147 should be better justified. As for the 3_6 pseudoknot the agreement with Cryo-EM data makes a good argument, the arguments for iFoldRNA and SimRNA seem rather weak.

6. More than once the authors present the transition from L shape to linear as important for the action of the frameshifting element. It should be made clear if this is a speculation put forward by the authors or if there is some evidence supporting it, other than the fact that this transition is observed in their simulations.

7. It would be useful to present more data from MD simulations in SI in order to evaluate their convergence and to appreciate the structural transitions discussed in the text. Several analysis tools specific for RNA are currently available that would produce nice plots to present MD data.

8. The choice of the simulation environment should be discussed, in particular with respect to ionic conditions that are well known to influence RNA structures as observed in chemical probing and reflected in simulations. The authors have used some "standard" simulation conditions. Is this choice appropriate with their SHAPE? Is the absence of MG++ justified?

Minor remark:

In figure 1 is "the 3_3 conformation contains an extra flanking stem SF" correct? Shouldn't it be the 3_6 conformation.

Reviewer #3 (Remarks to the Author):

The authors performed microsecond molecular dynamics simulations of regions of the SARS-CoV-2 mRNA relevant for frameshifting to investigate the mechanism of structural dynamics of this important RNA (specifically, the ORF1a/1b junction). Because the frameshifting element is conserved, it is an interesting therapeutic target (e.g., none of the many mutations encompassed by the omicron variant affect the frameshifting element). Thus, any elucidation of the mechanism will help provide new insights for therapeutic design. In terms of mechanism, frameshifting, in general, is not well understood. The predominant mechanism is the 'roadblock', where a structural element folds and alters ribosome movement along the mRNA. However, structural dynamics at both the secondary and tertiary levels are likely to play a role. The structural dynamics of mRNA is poorly understood. Molecular dynamics simulations, in combination with other experimental data, such as SHAPE probing and cryo-EM, is one of the few methods available for interrogating structural dynamics at the atomistic level. Using their previous SHAPE probing experiments and graph theory analysis as a basis, the authors explore several different secondary structures by taking the key step of studying three different RNA lengths (77, 87, and 144 nts), finding a high dependence of the conformation on the RNA length (e.g., length-dependent triplet formation and length-dependent stem formation). Namely, the length of the RNA can shift the equilibrium from one fold to another fold. This is an important result because length of the frameshifting element available to operate changes during the progression of translation by the ribosome. The authors characterize the molecular dynamics simulation trajectories with principal component analysis and also perform mutations designed to further probe the mechanism of the RNA, finding that mutations stabilize the 3₃ conformation. They suggest that the H-type pseudoknot in a ring conformation threaded by the 5'-end could promote ribosomal pausing. In addition, a transition between an L-conformation and a linear (i.e., vertical stacking of stems) conformation may be inhibited by their mutation. In the end, the authors identify structural features and motions that help determine frameshifting mechanisms. They suggest that axial bending from the L-conformation to the linear conformation produces fluctuations in mRNA tension that could promote frameshifting. Another key finding is the sensitivity of a pseudoknot (and therefore associated frameshifting) to junction nucleotides. The overall results are consistent with previous SHAPE probing experiments, cryo-EM experiments and crystallographic experiments, producing an integrated picture of the operation of this important RNA. This study is one of the first of its kind and offers relatively new, specific implications for antiviral strategies, including targeting threading, structural transitions, and pseudo-knot stabilizing interactions, with specific targets for small molecule design.

Comments

Overall, this is an important and timely contribution and deserves to be published in Nature Communications. I have several minor concerns.

1. In the results section, it is easy to 'get lost in the details'. There needs to be more explanatory sentences about what each group of interactions means, or what each motion means, interwoven throughout the results section.
2. Currently, microsecond-long trajectories are near the state-of-the-art in molecular dynamics simulation; however, this time scale does not necessarily constitute equilibrium. The authors need to comment on the limitations of microsecond sampling and discuss their methodology in the context of the many enhanced sampling techniques available today (e.g., Vaiana and Sanbonmatsu, JMB, 2009).
3. There are also some limitations of SHAPE probing data, their interpretation and their relation to RNA structure. These should be discussed.

4. Figures 1-2 – add L1, L2,L3 labels to the rainbow arc diagrams.

5. The work has important implications not just for COVID-19 but for viral mRNAs in general, and for human mRNAs. The authors should comment on the state-of-the-art in understanding mRNA mechanism in general, designing antivirals that target mRNAs in general and in engineering mRNAs for other diseases such as cancer.

6. Magnesium plays an important role in RNA folding, structure, dynamics and function. The authors should comment on this and on future potential studies that will include magnesium.

7. RNA studies typically use potassium rather than sodium. The authors should comment on their choice of sodium instead of potassium.

8. The strategy of studying RNAs of different lengths is a useful strategy in the case of examining mRNA-ribosome effects and for studying co-transcriptional effects. The authors should comment on the utility of their strategy to study RNAs with time-dependent lengths and the ubiquity of these systems.

Response to Reviewers for “*Length-dependent motions of SARS-CoV-2 frameshifting RNA pseudoknot and alternative conformations suggest avenues for frameshifting suppression*”

April 5, 2022

Manuscript ID: NCOMMS-21-49245

Title: “Length-dependent motions of SARS-CoV-2 frameshifting RNA pseudoknot and alternative conformations suggest avenues for frameshifting suppression”

Authors: Shuting Yan, Qiyao Zhu, Swati Jain, and Tamar Schlick

Dr. Nele Hug, Associate Editor
Nature Communications
nele.hug@springernature.com

Dear Dr. Hug,

Thank you for handling our paper and the reviewers for their thoughtful and constructive comments. Enclosed is our revision addressing the reviewers’ comments along with point by point responses.

In brief, we have addressed the experimental evidence for alternative FSE conformations (by SHAPE), assessment of the initial FSE conformations and trajectory validation, improvement of the results presentation, discussion of computational and modeling limitations, and other points raised. We have also amended some trajectories to be consistent with our validation criteria.

We hope you will find the revision acceptable for publication in *Nature Communications*.

Best wishes,

Tamar Schlick
Professor of Chemistry, Mathematics, and Computer Science
New York University

Response to Reviewer 1

The manuscript treats the novel idea that conserved SARS-CoV-2 RNA regions may be good drug targets, and in particular the frameshifting element (FSE). The authors derive three frameshift conformations and perform MD, leading to the conclusion that ring-like conformations may exist. The work is quite computationally thorough and may form a basis for the design of inhibitors. I recommend publication subject to the following aspects:

Reply: Thank you!

My main questions are as follows:

(a) There appears to be no experimental conformation of the alternative conformations found. Some discussion of how this could be achieved would be merited.

Reply: Thank you for the opportunity to clarify this point. Our alternative 2D conformations were found using SHAPE chemical probing data in combination with restrained 2D structure prediction (*JACS* 143: 11404, 2021); these experimental data are now highlighted more clearly in the Introduction and Discussion. Chemical reactivity experiments from other groups have also suggested additional alternative FSE conformations, such as 3_8 (kissing hairpin) by Huston et al. (*Mol. Cell*, 2021), and 2_2 (alternative Stem 1) by Sun et al. (*Cell*, 2021); see our extended discussion and Table 2 in our JACS paper on alternative conformations.

Currently, all available Cryo-EM and crystallographic structures correspond to the 3_6 motif. However, as we have shown in Schlick et al. *Biophys. J.* (2020) and *JACS* (2021), the FSE conformation is highly length dependent. To find 3_3, in fact, our computational analysis suggested using a 144-nt FSE sequence construct to probe experimentally. Indeed, the resulting SHAPE analysis for this length suggests that 3_3 dominates here. In addition, other SHAPE data for our predicted mutants yielded consistent reactivity data with our predictions: 3_3 for our 77-nt mutant [U4C, G71A, G72U], and 3_5 for 77-nt mutant [G72C, U74C].

(b) Is there any evidence for convergence of the MD with regards to the conformation space of the FSE.

Reply: The MD trajectory convergence is now examined by the evolution of system density (NPT ensemble), RMSD, eRMSD, radius of gyration, and hydrogen bonding (see Supplementary Figure S2-6 for wildtype systems and Figures S9-13 for mutants). All systems have steady density levels in the NPT ensemble. Those that had large RMSD fluctuations over the last 500 ns simulations were extended for another 250-500 ns, and reached plateaus. The eRMSD is a supplementary check for base interactions, measuring the distance between two 3D structures by considering the relative positions and orientations of their nucleobases. All simulations exhibit stable eRMSD evolutions over the last 500 ns. Similarly, steady states of radii of gyration and hydrogen bonds are satisfied. We then use these stable last 500 ns trajectories for all our analyses. Hence, our MD simulations can be considered convergent around a local region of FSE conformation space.

To sample the global conformation space, enhanced sampling methods are required, which we plan in future work. Nevertheless, the large number of systems modeled here, which start from different initial 3D predictions, provide a variety of candidate neighborhoods to help piece together the complex FSE conformation space. In Figure 2 of our paper (also attached below), we show a multi-trajectory clustering, where we extract 50 equally spaced frames from each of the four 77-nt 3_6 trajectories (200 frames in total) and perform PCA clustering, and likewise for the four 77-nt 3_5 trajectories. Note that we analyze 3_6 at 77-nt because this motif is thought to be dominant at this length. We use 77-nt for 3_5 too because this 3-way junction is only observed at this

length. For 3.3, which dominates at length 87-nt using 10 more upstream residues, no clustering is performed because only one trajectory maintains this motif throughout the MD simulations. The 3.6 and 3.5 trajectories form different clusters, but some are close to one another, such as trajectories 3 and 4 for 3.6 and trajectories 17 and 19 for 3.5, which sample similar conformations. Overall, we see a heterogeneous landscape, with parts revealed by our different trajectories.

Figure 2: Clustering analysis and representative structures for the three FSE motifs. Multi-trajectory clustering is performed to compare the 77-nt 3.6 trajectories and select representative systems, and the same for 3.5 trajectories, with 50 structures extracted from each. For 3.6, two representative trajectories are chosen, sampling an L and a linear shape, respectively. For 3.5, the two representative models capture compact and elongated shapes. For 87-nt 3.3, only one system maintains the motif throughout the MD simulation.

(c) Can the authors say something about the transition path between the conformations. PCA is a start for this but typically does not go all the way.

Reply: As we have discussed in Results (Figure 6), we have identified a conformation in the 3.3 77-nt trajectory that contains interactions present in all three FSE conformations (3.6, 3.3 and 3.5). Such an intermediate will be key in planned enhanced sampling simulations to capture associated transitions among these three-stem RNA states. This formidable research project will be pursued in the future on its own right. More on this was added to the Discussion, including an updated Figure 9 to show a proposed transition from 3.3 to 3.6 based on our MD models.

Editing for English quality would be useful.

Reply: We have edited the manuscript for clarity, especially the Results and Discussion sections.

Response to Reviewer 2

The manuscript by Schlick and co-authors presents the study of several putative structures of the SARS-CoV-2 frameshifting pseudoknot using graph theory and molecular dynamics to shed light on possible dynamic transition of this element and its role in translation of the viral genome. This element is an important therapeutical target since it regulates the expression of half of the viral proteins, and it is well conserved in all the known variants of the virus as well as in related viruses such as SARS-CoV. Moreover, the mechanism of frameshift is common to several other viruses and its mechanism is still largely unknown. Using graph theory to represent RNA secondary structures including pseudoknots the authors are able to identify three possible architectures of the 77-nt pseudoknot and of two larger systems containing the frameshifting sequence and additional portions of the sequences elongating the element.

The approach is interesting as the comparison of elements of different length can bring light to the dynamic transformations that the element undergoes as it is read by the ribosome, and therefore made shorter.

Reply: Thank you!

I have however important reservations on the results obtained mainly due to the choice of the initial configurations. The authors use several folding predictions software to build possible 3D structures for the three architectures found by graph theory. While the 2D structures are supported by chemical probing data, giving them some certainty on their actual existence, I could not find in the manuscript nor in the SI some convincing evidence that the predictions made for the 3D are plausible. Indeed, the different prediction tools propose significantly different structures for the various systems. The authors seem to use all these structures in their analysis (which in my view is a good thing), but in my opinion given the uncertainties in these initial structures the results should be presented with more caution and making more connection/comparisons with existing experimental data. Probably because of the large number of initial structures used, I often found myself lost in the result sections not knowing what was used when and what was a result of MD simulations. In its present state I think the result section is hard to follow, at least up to the discussion on mutations where things become clearer. My main suggestions would therefore be to clearly substantiate the initial structures giving the experimental evidence of their relevance and to revise the result section accordingly and also to make it easier to follow.

A few remarks in particular:

1. Can the authors compare the predicted initial structures against the SHAPE data used also in the 2D predictions? It is true that these data are already included in the prediction, but they have undergone several steps and approximations in making the 2D prediction. I think it would be worth to compare the secondary structures of the predicted 3D structures (both initial and after MD) to verify that the pairings predicted by SHAPE are actually there, especially for the possible 3D contacts which are harder to include in secondary structure predictions. I think that such an analysis would greatly strengthen the validity of the results. Possibly using also other kind of experimental data, it will then be useful to clearly state among the possible initial structures which are the ones supported by experimental evidence (and what) and which are purely putative.

Reply: Thank you for these excellent suggestions. Indeed, our intention as you state in considering many possible 3D models compatible with the 2D motifs is to survey the conformational space of the FSE broadly. This FSE structural landscape is likely heterogeneous and rugged, and a compendium of local sampling of each neighborhood can help piece together parts of a highly complex picture.

To address your point, we have now added a section “MD model validation and selection” to Results and Table S1-S3 to SI. We validated both our initial 3D predictions and the MD trajectories by comparing our models to SHAPE 2D structures and available experimental 3D structures. Representative systems were selected based on the validation tests, and we then focused on these representatives throughout the Results. Moreover, we have numbered the validated trajectories as shown in Table 1 in the paper and also here, and indicated the systems analyzed and plotted using these numbers.

For wildtype systems, we assessed the initial 3D predictions by reporting corresponding graph motifs and 2D structures (Table S1). For each model, the Hamming distance between the 2D structure and the SHAPE derived 2D structure was calculated. Models with wrong motifs or Hamming distances >10 were rejected, i.e., not considered for MD simulations.

The validated initial predictions were then subject to microsecond MD simulations. The start, middle, and end frames of the MD trajectories, as well as the cluster center structures, were again validated using graph motifs, 2D structures, and 3D clashes (Table S2). “Clashscores” were calculated as the number of serious clashes per 1000 atoms using MolProbity. Models having wrong motifs were rejected, and those having Hamming distances >10 or clashscores >5 were noted, making them less likely to be chosen as representatives. All validation testing are summarized in Table S3 (also shown on the next page). As we see, 3 models were rejected during the initial prediction validation step, and 4 models were rejected during the MD trajectory validation step.

We have also aligned our 3.6 models with the 4 available experimental structures in Table S2 (two Cryo-EM structures by Zhang et al. and Bhatt et al., and two crystallographic structures by Roman et al. and Jones et al.), and we show the best alignments here in Table S3. Our iFoldRNA models align well with the crystal structures (3-4 Å RMSD), and our RNAComposer models align reasonably with the Cryo-EM structures (10-11 Å RMSD), lending confidence in our modeled 3.6 conformations.

For mutant systems, similar protocols were followed. Correct motifs and consistent 2D structures with SHAPE were maintained. Instead of using the Hamming distance to evaluate the 2D structures, we tracked the Stem 2 lengths over the trajectories (Table S5, also shown below). Because the three FSE motifs differ only by Stem 2, this stem length is a measure of the motif strength. Most mutant systems have longer Stem 2 than the wildtype by design, and those with longest Stem 2 were chosen as representatives.

All FSE systems considered ⁽³⁷⁾ and validated ⁽³⁰⁾						
Length	WT 3.6	WT 3.3	WT 3.5	M 3.6	M 3.3	M 3.5
77-nt	R ¹ , S ² , I ³ , V ⁴	R, S ¹⁰ , I ¹¹ , V ¹²	R ¹⁶ , S ¹⁷ , I ¹⁸ , V ¹⁹	R ^{1'} , S ^{2'} , I ^{3'} , V ^{4'}	I ^{11'}	R ^{16'} , S ^{17'} , I ^{18'} , V ^{19'}
87-nt	R ⁵ , S, I ⁶ , V	R, S, I ¹³ , V	N/A	N/A	N/A	N/A
144-nt	R ⁷ , I ⁸ , F ⁹	R, I ¹⁴ , F ¹⁵	N/A	R ^{7'} , I ^{8'}	N/A	N/A

Table 1: For each motif (3.6, 3.3, and 3.5) and length (77, 87, and 144-nt) combination, we use five 3D prediction programs (R for RNAComposer, S for SimRNA, I for iFoldRNA, V for Vfold3D, and F for Farfar2) to generate starting models. Models rejected initially are colored light blue, and those rejected during the MD trajectory are shown in brown. The 19 convergent and validated wildtype MD trajectories and the 11 mutant trajectories are all numbered for reference in the superscript. The representative trajectories are highlighted yellow.

Rejected Models			
Conformer	Program	Step	Rejection reason
87-nt 3_6	SimRNA	Initial	Wrong motif, Hamming 14
87-nt 3_3	SimRNA	Initial	Hamming 14
144-nt 3_3	RNAComp	Initial	Wrong motif, Hamming 16
87-nt 3_6	Vfold3D	MD	Wrong motif (cluster)
77-nt 3_3	RNAComp	MD	Wrong motif (MD mid, end, cluster)
87-nt 3_3	RNAComp	MD	Wrong motif (MD start, mid, end, cluster)
87-nt 3_3	Vfold3D	MD	Wrong motif (MD mid, end, cluster)

Accepted Models				
Conformer	Program	Warnings	Crystal RMSD (Å)	Cryo-EM RMSD (Å)
77-nt 3.6	1 RNAComp	1	5.50 (Roman)	10.93* (Zhang)
	2 SimRNA	3	13.47 (Roman)	11.48 (Bhatt)
	3 iFoldRNA	0	3.67 (Jones)	13.00 (Bhatt)
	4 Vfold3D	0	3.65* (Jones)	12.68 (Bhatt)
87-nt 3.6	5 RNAComp	0	7.79 (Jones)	9.88* (Bhatt)
	6 iFoldRNA	3	3.99* (Jones)	14.25 (Zhang)
144-nt 3_6	7 RNAComp	3	4.17 (Jones)	11.35 (Zhang)
	8 iFoldRNA	1	3.55* (Jones)	12.31 (Zhang)
	9 Farfar2	3	4.10 (Roman)	10.57* (Zhang)
77-nt 3.3	10 SimRNA	3		
	11 iFoldRNA	0		
	12 Vfold3D	2		
87-nt 3.3	13 iFoldRNA	1		
144-nt 3_3	14 iFoldRNA	3		
	15 Farfar2	3		
77-nt 3.5	16 RNAComp	3		
	17 SimRNA	3		
	18 iFoldRNA	0		
	19 Vfold3D	0		

Table S3: Summary table for the wildtype model validations. For rejected models, we specify step number (initial or MD validation) and reason for rejection. For accepted models, we specify how many warnings they receive due to large Hamming distances and high clashscores. In addition, we list the best alignment RMSDs between our 3.6 cluster centers and the two crystal structures by Jones et al. (PDB ID: 7LYJ) and Roman et al. (PDB ID: 7MLX), as well as the best alignment with the Cryo-EM structure by Zhang et al. (PDB ID: 6XRZ) and the FSE segment extracted from the Bhatt et al. mRNA-ribosome Cryo-EM complex (PDB ID: 7O7Z). The lowest RMSDs are labeled with asterisk for each length. Trajectory numbers 1-19 refer to labels used in Table 1 with representatives highlighted in yellow.

Conformer	Program		Stem 2 base pairs				
			Initial	MD start	MD mid	MD end	Cluster
77-nt M3.6	1'	RNAComp	9	9	4	4	4
	2'	SimRNA	9	9	8	9	9
	3'	iFoldRNA	8	9	6	7	7
	4'	Vfold3D	9	9	9	8	8
144-nt M3.6	7'	RNAComp	8	8	4	5	5
	8'	iFoldRNA	5	5	4	5	4
77-nt M3.3	11'	iFoldRNA	7	7	6	7	7
77-nt M3.5	16'	RNAComp	6	7	7	6	7
	17'	SimRNA	6	7	7	7	7
	18'	iFoldRNA	6	7	7	7	7
	19'	Vfold3D	7	6	6	6	6

Table S5: Mutant FSE model validation. We monitor the Stem 2 length in the initial 3D model, MD start, MD middle, and MD end frames, as well as the largest cluster center structure. Trajectory numbers refer to labels used in Table 1. Models with longest (i.e., more stable) Stem 2 are chosen as representatives and highlighted in yellow.

2. Always on the choice of the initial structures, not all prediction tools have the same level of detail as some of them are coarse grained. However, the proposed structures are used assuming the same validity for all predictions, which is debatable. For the models coming from coarse graining an additional validation of the relevance of the proposed structure should be presented.

Reply: Thank you for this point as well. Indeed, the 3D prediction programs use different strategies: RNAComposer, Vfold3D, and Farfar2 use template-based fragment assembly; iFoldRNA and SimRNA produce coarse-grained RNA models that are sampled via Discrete Molecular Dynamics and replica exchange Monte Carlo, respectively.

Using our updated validation criteria for initial predictions and MD models, we found that many of the coarse-grained SimRNA models were rejected, or led to warnings for 2D structure inconsistencies and 3D structure clashes (Table S3). We thus mainly use RNAComposer and iFoldRNA as representative models in our work (Table 1).

The coarse-grained iFoldRNA models actually perform well in our validation tests. For the 3.6 pseudoknot, iFoldRNA models align well with the crystal structures that also possess a linear shape (3-4 Å RMSD), lending confidence to this model. For 3.3, the iFoldRNA model is the only system that successfully maintains the correct motif throughout the MD simulations at all three lengths. For 3.5, the iFoldRNA model has consistent 2D structure with the SHAPE input and low 3D clashes (Table S3). Moreover, using our multi-trajectory clustering analysis, we found that the iFoldRNA and the Vfold3D 3.5 models highly resemble each other (Figure 2 in the paper and shown on page 3), adding further confidence to these representations.

3. Part of the discussion focuses on the threading of a single strand through a ring formed by the pseudoknot. Again, I think that the existence of this threading should be substantiated with more experimental evidence. If clear evidence for this configuration exists it would be useful to spend a few words detailing it.

Reply: Yes, as we have discussed in connection with Figure 5 in the paper, the two Cryo-EM structures by Zhang et al. (*Nat. Struct. Mol. Biol.* 28: 747, 2021) and Bhatt et al. (*Science* 372: 1306, 2021) also show the 5' end threading in the 3_6 structure. Their Cryo-EM structures are shown and compared to our 3_6 MD models in Figure 5. We now highlight this point in the Introduction, Results under subsection “Threaded L shape vs. non-threaded linear shape”, and Discussion.

4. When presenting the three possible architectures, the authors discuss which one is dominant over the others depending to the length of the sequence. However, I could not find how this dominance is assessed/calculated. Does it have to do with the partition function in the 2D predictions? This point should be clarified.

Reply: [redacted] discussed extensively the length-dependent aspects of the FSE. In brief, the 77-nt (without the slippery site) is largely limited to the 3_6 conformation, but as length increases, competing interactions permit alternative hydrogen bonding networks and allow other stems to form. [redacted] illustrates that the FSE favors the 3_3 conformation over 3_6 when the sequence increases to 144-nt.

For this landscape estimate, we used the Gibbs free energy from the ShapeKnots 2D structure ensemble predictions, which incorporate SHAPE reactivity data as restraints. We then applied Boltzmann weighting $w_i = e^{-E_i/k_B T}$ for each structure i , and calculated the partition function $Z = \sum_i w_i$ and the probability $p_i = w_i/Z$. To find the probability of a certain conformation (3_6, 3_3, 3_5, etc.), we sum up probabilities of all the 2D structures that correspond to that conformation. We now briefly mention our conformational landscape approach in the Introduction.

[redacted]

5. The choice of the three representative models on page 3, 139-147 should be better justified. As for the 3.6 pseudoknot the agreement with Cryo-EM data makes a good argument, the arguments for iFoldRNA and SimRNA seems rather weak.

Reply: In our original version of our paper, we presented many models, but found this to be confusing. Thus, we now display representative models, as we detailed in the beginning of Results. Namely, for 3.6, we use RNAComposer, as you say, due to good agreement with the Cryo-EM structures. Now, we also performed multi-trajectory clustering analysis to find similarity and differences among the various 3.6 models. We found that the 3.6 RNAComposer model represents the L shape, and the 3.6 iFoldRNA model represents the linear shape. Moreover, the iFoldRNA model aligns well with the two crystal structures (3-4 Å RMSD). Hence, we also included the iFoldRNA models as representative structures for 3.6.

For 3.3, we chose iFoldRNA because only this system successfully maintained the 3.3 motif throughout the MD simulations at all lengths, and they have the best 2D structure agreement with SHAPE data overall (Table S3 shown on page 6).

For 3.5, following our extensive validation tests, we now use Vfold3D and iFoldRNA as representative models, as these two models produce better 2D structure agreement with SHAPE data and better MolProbity clash-score evaluations (Table S3). Moreover, using multi-trajectory clustering analysis, we found that Vfold3D and iFoldRNA models represent an elongated and a compact 3.5 structure, respectively (Figure 2). See also our response to Reviewer 1, point (b) and Figure 2 shown here on page 3.

All relevant figures and text in the Results were updated for these representative structures (Fig. 2-8).

6. More than once the authors present the transition from L shape to linear as important for the action of the frameshifting element. It should be made clear if this is a speculation put forward by the authors or if there is some evidence supporting it, other than the fact that this transition is observed in their simulations.

Reply: Thank you for highlighting this point. Yes, this transition is speculative and based on our PCA. Interestingly, both Cryo-EM structures by Zhang et al. (*Nat. Struct. Mol. Biol.* 28: 747, 2021) and Bhatt et al. (*Science* 372: 1306, 2021) exhibit the L shape, while the two crystal structures by Roman et al. (*ACS Chem. Biol.* 16: 1469, 2021) and Jones et al. (*RNA* 28: 239, 2022) display the linear shape. The crystal structure authors suggest that the FSE is highly dynamic before the ribosome approaches, and that it may adopt and switch among multiple conformations including the L and linear shapes. We have added this to the Results subsection "L to linear shape transitions in 3.6".

7. It would be useful to present more data from MD simulations in SI in order to evaluate their convergence and to appreciate the structural transitions discussed in the text. Several analysis tools specific for RNA are currently available that would produce nice plots to present MD data.

Reply: Thank you for suggesting this analysis to us. The MD trajectory convergence is now examined by the evolution of system density (NPT ensemble), RMSD, eRMSD, radius of gyration, and hydrogen bonding (see Supplementary Figure S2-6 for wildtype systems and Figures S9-13 for mutants). All systems have steady density levels in the NPT ensemble. Those that had large RMSD fluctuations over the last 500 ns simulations were extended for another 250-500 ns, and reached plateaus. The eRMSD is a supplementary check for base interactions, which measures the distance between two 3D structures by considering the relative positions and

orientations of their nucleobases, and all simulations exhibit stable eRMSD evolutions over the last 500 ns. Similarly, steady states of radii of gyration and hydrogen bonds are satisfied. Hence, our MD simulations can be considered locally convergent around a region of FSE conformation space.

8. The choice of the simulation environment should be discussed, in particular with respect to ionic conditions that are well known to influence RNA structures as observed in chemical probing and reflected in simulations. The authors have used some “standard” simulation conditions. Is this choice appropriate with their SHAPE? Is the absence of MG++ justified?

Reply: The folding buffer in our SHAPE experiments contained KCl and MgCl₂. We had concerns about using Mg²⁺ in our simulations, because Mg²⁺ placement could bias the MD. From the literature, it has been suggested that Mg²⁺ ions are not required for the refolding of the FSE pseudoknot, although they increase pseudoknot mechanical rigidity (Neupane et al., *Nat. Commun.* 12: 4749, 2021).

Our MD simulations without Mg²⁺ were performed so as not to bias the RNA conformations. Currently, there is only one experimental structure with Mg²⁺ bounded to the FSE available, reported by Jones et al. (*RNA* 28: 239, 2022). This crystal structure is a linear shape 3_6 pseudoknot, with a slightly modified RNA sequence, so it requires further examination before using it as a guidance to place Mg²⁺ in our predicted systems.

We have now commented on this in the Discussion. We definitely plan to focus on the divalent ion placement in our next FSE work.

Minor remark: In figure 1 is “the 3_3 conformation contains an extra flanking stem SF” correct? Shouldn’t it be the 3_6 conformation.

Reply: The flanking stem SF (gray) in Figure 1 is for 3_3 as we wrote. This flanking stem is an extra stem that appears in the 3_3 conformation once the sequence length reaches 87-nt or more, where the 5’ and 3’ ends base pair to enclose the 3 stems of 3_3.

Response to Reviewer 3

The authors performed microsecond molecular dynamics simulations of regions of the SARS-CoV-2 mRNA relevant for frameshifting to investigate the mechanism of structural dynamics of this important RNA (specifically, the ORF1a/1b junction). Because the frameshifting element is conserved, it is an interesting therapeutic target (e.g., none of the many mutations encompassed by the omicron variant affect the frameshifting element). Thus, any elucidation of the mechanism will help provide new insights for therapeutic design. In terms of mechanism, frameshifting, in general, is not well understood. The predominant mechanism is the “roadblock”, where a structural element folds and alters ribosome movement along the mRNA. However, structural dynamics at both the secondary and tertiary levels are likely to play a role. The structural dynamics of mRNA is poorly understood. Molecular dynamics simulations, in combination with other experimental data, such as SHAPE probing and cryo-EM, is one of the few methods available for interrogating structural dynamics at the atomistic level. Using their previous SHAPE probing experiments and graph theory analysis as a basis, the authors explore several different secondary structures by taking the key step of studying three different RNA lengths (77, 87, and 144 nts), finding a high dependence of the conformation on the RNA length (e.g., length-dependent triplet formation and length-dependent stem formation). Namely, the length of the RNA can shift the equilibrium from one fold to another fold. This is an important result because length of the frameshifting element available to operate changes during the progression of translation by the ribosome. The authors characterize the molecular dynamics simulation trajectories with principal component analysis and also perform mutations designed to further probe the mechanism of the RNA, finding that mutations stabilize the 3.3 conformation. They suggest that the H-type pseudoknot in a ring conformation threaded by the 5'-end could promote ribosomal pausing. In addition, a transition between an L-conformation and a linear (i.e., vertical stacking of stems) conformation may be inhibited by their mutation. In the end, the authors identify structural features and motions that help determine frameshifting mechanisms. They suggest that axial bending from the L-conformation to the linear conformation produces fluctuations in mRNA tension that could promote frameshifting. Another key finding is the sensitivity of a pseudoknot (and therefore associated frameshifting) to junction nucleotides. The overall results are consistent with previous SHAPE probing experiments, cryo-EM experiments and crystallographic experiments, producing an integrated picture of the operation of this important RNA. This study is one of the first of its kind and offers relatively new, specific implications for antiviral strategies, including targeting threading, structural transitions, and pseudo-knot stabilizing interactions, with specific targets for small molecule design.

Comments

Overall, this is an important and timely contribution and deserves to be published in Nature Communications. I have several minor concerns.

Reply: Thank you very much for your favorable comments.

1. In the results section, it is easy to “get lost in the details”. There needs to be more explanatory sentences about what each group of interactions means, or what each motion means, interwoven throughout the results section.

Reply: We have rewritten the Results section to make it more readable. We have also added subsection headings to highlight the main points in each paragraph or group of paragraphs.

2. Currently, microsecond-long trajectories are near the state-of-the-art in molecular dynamics simulation; however, this time scale does not necessary constitute equilibrium. The authors need to comment on the limitations of microsecond sampling and discuss their methodology in the context of the many enhanced sampling techniques available today (e.g., Vaiana and Sanbonmatsu, JMB, 2009).

Reply: Thanks for the opportunity to elaborate on this point. We use MD simulations here to relax the predicted models from various 3D prediction programs and locally sample the conformational space. We agree that microsecond simulations cannot explore the global conformational space. We have commented on this limitation in the Discussion. See also our response to Reviewer 1, point (b) and Figure 2 shown on page 3. In the future, we plan to pursue enhanced sampling to probe conformational transitions among the alternative FSE motifs.

3. There are also some limitations of SHAPE probing data, their interpretation and their relation to RNA structure. These should be discussed.

Reply: Indeed, SHAPE data can only help infer nucleotide reactivities rather than directly provide 2D structures. We have added this discussion of SHAPE usage/limitations to the Discussion.

4. Figures 1-2 – add L1, L2,L3 labels to the rainbow arc diagrams.

Reply: Thank you for the suggestion. We have added these loop labels to Figures 1-2.

5. The work has important implications not just for COVID-19 but for viral mRNAs in general, and for human mRNAs. The authors should comment on the state-of-the-art in understanding mRNA mechanism in general, designing antivirals that target mRNAs in general and in engineering mRNAs for other diseases such as cancer.

Reply: Thank you for highlighting this important point. We have now added this to the Discussion.

6. Magnesium plays an important role in RNA folding, structure, dynamics and function. The authors should comment on this and on future potential studies that will include magnesium.

Reply: The folding buffer in our SHAPE experiments contained KCl and MgCl₂. We had concerns about using Mg²⁺ in our simulations, because Mg²⁺ placement could bias the MD. From the literature, it has been suggested that Mg²⁺ ions are not required for the refolding of the FSE pseudoknot, although they increase pseudoknot mechanical rigidity (Neupane et al., *Nat. Commun.* 12: 4749, 2021).

Our MD simulations without Mg²⁺ were performed so as not to bias the RNA conformations. Currently, there is only one experimental structure with Mg²⁺ bounded to the FSE available, reported by Jones et al. (*RNA* 28: 239, 2022). This crystal structure is a linear shape 3_6 pseudoknot, with a slightly modified RNA sequence, so it requires further examination before using as a guidance to place Mg²⁺ in our predicted systems.

We have now commented on this in the Discussion. We definitely plan to focus on the divalent ion placement in our next FSE work.

7. RNA studies typically use potassium rather than sodium. The authors should comment on their choice of sodium instead of potassium.

Reply: We examined the effect of potassium on the FSE structure using the 77-nt 3_6 RNAComposer system as a case study (system neutralized with potassium ions and the same 0.1 M concentration of KCl added). The largest cluster center structures from the MD trajectories with potassium and sodium are highly similar (see Figure below). We plan to explore the ion issue thoroughly in our future FSE work.

For Reviewers Figure 1: MD screenshots of 77-nt 3_6 structure by RNAComposer with Na^+ (purple) and K^+ (pink).

8. The strategy of studying RNAs of different lengths is a useful strategy in the case of examining mRNA-ribosome effects and for studying co-transcriptional effects. The authors should comment on the utility of their strategy to study RNAs with time-dependent lengths and the ubiquity of these systems.

Reply: Thank you for the suggestion. We have now added this to the Discussion.

REVIEWERS' COMMENTS

Reviewer #1 (Remarks to the Author):

My comments have been addressed and the manuscript should be published.

Reviewer #2 (Remarks to the Author):

The authors have successfully addressed all my concerns.

The results section is now much clearer, and many links with experimental data have been drawn, making results much more solid than what was presented initially.